

# Beam tracking strategies for fast acquisition of solar wind velocity distribution functions with high energy and angular resolutions

Johan De Keyser[1], Benoit Lavraud[2], Lubomir Přech[3], Eddy Neefs[1], Sophie Berkenbosch[1], Bram Beeckman[1], Andrei Fedorov[2], Maria Federica Marcucci[4], and Daniele Brienza[4]

[1]Royal Belgian Institute for Space Aeronomy (BIRA-IASB), Ringlaan 3, B-1180 Brussels, Belgium
[2]Institut de Recherche en Astrophysique et Planétologie (IRAP), Université de Toulouse, CNRS, UPS, CNES, Toulouse, France
[3]Charles University, Faculty of Mathematics and Physics, Prague, Czech Republic
[4]Istituto di Astrofisica e Planetologia Spaziali (INAF/IAPS), Rome, Italy

**Correspondence:** J. De Keyser
(Johan.DeKeyser@aeronomie.be)

**Abstract.** Space plasma spectrometers have often relied on spacecraft spin to collect three-dimensional particle velocity distributions, which simplifies the instrument design and reduces its resource budgets, but limits the velocity distribution acquisition rate. This limitation can in part be overcome by a the use of electrostatic deflectors at the entrance of the analyser. By mounting such a spectrometer on a sun-pointing spacecraft, solar wind ion distributions can be acquired at a much higher rate because

5    the solar wind ion population, which is a cold beam that fills only part of the sky around its mean arrival direction, always remains in view. The present paper demonstrates how the operation of such an instrument can be optimized through the use of beam tracking strategies. The underlying idea is that it is much more efficient to cover only that part of the energy spectrum and those arrival directions where the solar wind beam is expected to be. The advantages of beam tracking are a faster velocity distribution acquisition for a given angular and energy resolution, or higher angular and energy resolution for a given acqui-

10    sition rate. It is demonstrated by simulation that such beam tracking strategies can be very effective while limiting the risk of losing the beam. They can be implemented fairly easily with present-day on-board processing resources.

**Keywords.** Interplanetary physics (instruments and techniques, solar wind plasma)

## 1    Introduction

15    The plasma in the outer layers of the solar atmosphere is so hot that even the sun's gravity cannot restrain it. The sun therefore produces a persistent stream of plasma that flows almost radially away in all directions. This "solar wind" consists of electrons and ions (protons with a limited admixture of alpha particles and trace amounts of highly ionised heavier elements) and constitutes an overall electrically neutral plasma. The solar wind can be regarded as a turbulent medium that is driven by



free energy from the differential motion of plasma streams that cascades via Alfvén waves down to kinetic scales where it is dissipated (e.g. Coleman, 1968; Tu and Marsch, 1995; Bruno and Carbone, 2005). Studies of solar wind turbulence at kinetic scales require the acquisition of full three-dimensional velocity distribution functions (VDFs) with high energy resolution and high angular resolution at a rapid cadence to be able to observe various signatures of the underlying processes in the VDFs (e.g.

Marsch, 2006, 2012; Kiyani et al., 2015; Valentini et al., 2016), while maintaining a sufficient signal-to-noise ratio. Achieving all these objectives at the same time is a daunting task that places stringent performance requirements on plasma spectrometer hardware.

On early solar wind missions such as Helios 1 and 2 (Porsche, 1981), where the satellite spin axis was perpendicular to the ecliptic, the plasma instruments actively scanned over energy by rapidly stepping the analyser potential, simultaneously

measuring over a range of angles in the plane containing the spin axis, while scanning over angles in the plane perpendicular to the spin axis with spacecraft rotation (Rosenbauer et al., 1977, 1981). The spacecraft spin rate ($60\,\mathrm{s}$ in this case) is the maximum solar wind VDF time resolution that can be achieved with such a setup, unless multiple instrument heads are installed (as has been done, for instance, for the Fast Plasma Investigation instruments on NASA's Magnetospheric MultiScale spacecraft (Pollock et al., 2016)). A similar situation occurs on the Cluster satellites (Escoubet et al., 2001). Their $4\,\mathrm{s}$ spin period thus

leads to a correspondingly better time resolution for solar wind measurements with the CIS-HIA instrument (Rème et al., 2001). To do even better, one must ensure that the solar wind always remains in the field of view of the detector. This can be achieved with a 3-axis stabilized platform (e.g., Solar Orbiter (Müller et al., 2013)) or with a spinning spacecraft that has its spin axis pointing toward the Sun (e.g. as was proposed for THOR (Vaivads et al., 2016)). An instrument that always looks at the Sun, however, must create a VDF by sampling different energies and directions simultaneously by using multiple detectors

or by actively scanning over energies and directions, or a combination of both. For example, the BIFRAM spectrometer on Prognoz 10 used a hybrid approach, with multiple analysers simultaneously sampling along the Sun–Earth line and scanning over energy in a time-shifted way to obtain a $63\,\mathrm{ms}$ time resolution, and at the same time using several detectors pointing from $7°$ to $24°$ away from the solar direction along different azimuth angles; while not covering the full sky, combining these data leads to representative energy spectra with a time resolution of $640\,\mathrm{ms}$ (Vaisberg et al., 1986; Zastenker et al., 1989), a rate

much faster than the spacecraft spin ($118\,\mathrm{s}$). Another approach is to have multiple detectors over only one angular coordinate (azimuth) but to scan actively over energy and the other angle (elevation). This can be implemented by placing a deflector system in front of the spectrometer entrance, as has been done for SWA-PAS on Solar Orbiter (Marsden and Müller, 2011) and as has been envisaged for the THOR-CSW ion spectrometer (Cara et al., 2017). Such instruments need a high geometric factor to ensure an appropriate signal-to-noise ratio even with short exposure times. Short exposures are a necessity if the full VDF

must be obtained rapidly, especially if the number of energy and elevation bins is high.

To meet these requirements, a variety of technologies must be considered, not only to build the instrument but also to operate it. In the present paper we address techniques for selectively sampling the energy and angular bins so as to cover only those voxels in energy–elevation–azimuth space where the solar wind beam is expected to be found. Indeed, at any given time only a fraction of all possible energy–elevation–azimuth voxels contain a significant number of particles. It is therefore

natural to sample the solar wind beam only around the expected energy and orientation, a process called "beam tracking".



**Table 1.** Solar wind parameters at $1\,\mathrm{au}$ and instrument requirements

| Parameter | slow wind | fast wind |
|---|---|---|
| Speed [km·s$^{-1}$] | 350 | 800 |
| ICME speed [km·s$^{-1}$] | < 2000 | < 2000 |
| Shock speed jumps [km·s$^{-1}$] | < 200 | < 200 |
| Proton thermal speed [km·s$^{-1}$] | 20–40 | 40–80 |
| Tangential speed jumps [km·s$^{-1}$] | < 80 | < 80 |
| Energy range [eV] | 640 | 3330 |
| ICME maximum energy [eV] | < 20000 | < 20000 |
| Shock energy jumps [eV] | 900 | 1900 |
| Proton thermal energy [eV] | 2–8 | 8–33 |
| *Required energy range* [eV] | 600 to 20000 | |
| *Minimum energy window*[a] | 5 % | |
| *Recommended energy window*[a] | 20–30 % | |
| Solar wind aberration[b] [°] | 3–7 | 1–3 |
| Range of direction [°] | −13 to +13 | −6 to +6 |
| Thermal beam width [°] | −7 to +7 | −6 to +6 |
| *Required angular range* [°] | −24 to +24 | |
| *Minimum angular window*[a] [°] | 24 | |

[a] Windows are computed between ±2 standard deviations.

[b] The solar wind aberration is the angle between the apparent solar wind direction
and the Earth–Sun line and is 3° on average. It is assumed that the instrument axis is
pointing toward the aberrated solar wind direction to within a few degrees.

The purpose of this paper is to examine beam tracking strategies for electrostatic plasma analysers. Both energy tracking and angular tracking are considered (section 2). We describe how these strategies can be implemented (section 3). The performance of these strategies is then tested in section 4 with synthetic data, some of which are based on actual high-cadence solar wind data. A summary of the capabilities of beam tracking techniques and an outlook on other domains in which they can be applied
5  is presented in section 5.

## 2  Beam tracking

Plasma spectrometers build up a VDF by detecting particles while scanning through three-dimensional velocity space. Plasma spectrometers typically gauge particles using an energy filter in the form of a quadrispheric electrostatic analyser (e.g. Carlson et al., 1982; Bame et al., 1992; Rème et al., 2001), although some new designs are emerging (e.g. Bedington et al., 2015; Skoug
10  et al., 2016; Morel et al., 2017). Specifically relevant for beam tracking applications are spectrometers where an electrostatic





elevation filter (using a transverse electric field set up between converging deflection plates) is placed in front of the analyser (e.g. McComas et al., 2007; Cara et al., 2017). Measurements are made over a range of azimuths simultaneously with a segmented anode array at the exit of the analyser. The particles are detected by means of a micro-channel plate or by channeltrons, each of which has its own advantages and drawbacks.

Since the solar wind is usually supersonic and even super-Alfvénic (with rare exceptions, Chané et al., 2015), the solar wind thermal velocity (usually several 10s of $\mathrm{km \cdot s^{-1}}$ is well below the bulk velocity. In addition, the thermal energy is much less than the range of variation of the beam energies corresponding to slow and fast solar wind (see, e.g., Gosling et al., 1971; McComas et al., 2000, 2002). The solar wind speed vector can vary by $500\ \mathrm{km \cdot s^{-1}}$ and more near interplanetary shocks, and can reach up to $1500\,\mathrm{km \cdot s^{-1}}$ and more in interplanetary coronal mass ejections (e.g. Gosling et al., 1968; Volkmer and

Neubauer, 1985; Dryer, 1994; Watari and Detman, 1998; Wu et al., 2016). Such dramatic changes occur over seconds to many minutes. The speed vector can change tangentially in solar wind discontinuities by $> 100\,\mathrm{km \cdot s^{-1}}$ (see, e.g., Borovsky, 2012; Borovsky and Steinberg, 2014; Burlaga, 1969; De Keyser et al., 1998); the jump is below $\sim 65\,\mathrm{km \cdot s^{-1}}$ in $99\,\%$ of the cases. Table 1 summarizes the implications of these numbers for the energies and solar wind arrival angles at $1\,\mathrm{au}$ (for a comparable exercise for heliocentric distances down to $0.23\,\mathrm{au}$, see McComas et al., 2007). It is clear that the solar wind beam typically

occupies only part of the energy–elevation–azimuth space that the instrument must be able to handle.

    Beam tracking consists in making a prediction about the energy and orientation of the solar wind beam before one starts a VDF measurement. Such a prediction may be obtained from the preceding measurements of the instrument itself, or may be based on data provided by other instruments (e.g. Faraday cup detectors) that can produce ion moment data at an even higher cadence; here the two variants are called "internal" and "external" beam tracking, respectively. Based on that prediction, the

energy and angular windows can be defined over which the spectrometer has to scan to obtain the next VDF with minimum effort.

## 2.1   Energy tracking

The energy range is essentially determined by the solar wind speed range and must go from $< 600\,\mathrm{eV}$ to $\sim 20\,\mathrm{keV}$. The width of the energy window must cover at least 4 times the thermal proton energy. Since the energy range is usually discretised

logarithmically (see below), the beam width should be at least $15\,\%$ of the full log-energy range. However, a more stringent requirement is that the energy window must be wide enough to avoid losing the solar wind beam upon sudden changes; depending on the VDF acquisition cadence, a width of $20$–$30\,\%$ of the full log-energy range seems to be a reasonable choice as will be shown below.

    The transmission properties of such an electrostatic analyser are such that only particles within a specified energy range

$\delta E$ are able to reach the detector, with a constant $\Delta E/E$ defining the energy resolution of the instrument. It is therefore natural to divide the energy range logarithmically into $N_E$ bins. A typical solar wind measurement does not necessarily have to scan all those bins, but may be limited to a number $N_E^* \leq N_E$ corresponding to the energy window width derived above. "Energy tracking" then refers to intelligently choosing the bins that have to be scanned so that no significant parts of the energy distribution are left unsampled.



The total number of energy bins is fixed by the energy range to be covered, and by the energy resolution one wants to achieve, by

$$N_E = \frac{\log E_{\mathrm{max}} - \log E_{\mathrm{min}}}{\delta E/E}.$$

When performing energy tracking over an energy window of $\Delta E$, the number of energy bins to be sampled is only

$$5 \quad N_E^* = \frac{\log(E + \Delta E/2) - \log(E - \Delta E/2)}{\delta E/E} \approx \frac{\Delta E}{\delta E}.$$

In general, one can choose both the centre of the energy window that has to be scanned and the width of that window. Changing the width of the window could be a way to take into account the changing temperature of the solar wind. It is, however, not recommended to do this. First, such decisions have to be made on-board and very fast, and deciding on the window width might be quite difficult if the VDFs have complicated shapes. Second, as discussed above, the width of the window 10 is mostly determined by the need to handle rapid time variations. A third drawback is that this would make the duration of VDF acquisition variable and thus unpredictable, which usually is considered undesirable from the point of view of on-board instrument management. This also is impractical for data handling.

Usually the VDF sampling is centred on the mean energy. Alternatively, it is possible to systematically shift the energy window upwards from the mean proton energy to minimise the chances of missing the peak of the $He^{++}$ contribution, which 15 for the same mean velocity has an energy-over-charge that is twice that of the dominant proton population; in such an $\alpha$-particle operating mode, the number of energies in a scan has to be large enough to include the proton and $\alpha$ peaks with sufficient margin (for THOR-CSW design, $N_E^* \geq 24$, so that the energy range spans at least a factor of $5.6$).

## 2.2 Angular tracking

A similar reasoning applies to the angular range of the solar wind beam. The thermal beam width suggests a minimum sampling 20 width of $24°$, centred around a solar wind arrival direction that can vary within a certain range around the average aberrated solar wind direction (Fairfield, 1971), as indicated in Table 1. In general, $N_\theta^* \leq N_\theta$ and $N_\alpha^* \leq N_\alpha$ for elevation and azimuth, respectively.

The use of wider windows may help to avoid missing temperature anisotropy effects in the VDFs (Marsch et al., 2006; Marsch, 2012) or the presence of suprathermal beams and/or extended plateaus in the VDFs (Marsch et al., 2009; Osmane et al., 25 2010; Marsch, 2012; Voitenko and Pierrard, 2013). Beam tracking strategies follow essentially the core of the distribution. In order not to miss features that may appear outside of the thermal wind advection cone, the actual energy–elevation–azimuth windows selected for data acquisition must be large enough.

## 2.3 Theoretical speed-up

Scanning the complete set of energies, elevations, and azimuths requires a time

$$30 \quad \Delta t_{\mathrm{full}} = N_E N_\theta N_\alpha \delta t / N_{\mathrm{par}}$$



where $\delta t$ is the time needed for accumulating particle detections in a single energy–elevation–azimuth bin, and $N_{\mathrm{par}}$ is the number of bins that are sampled simultaneously. In the THOR-CSW design, for instance, all azimuths are sampled in parallel by having a dedicated anode for each azimuth, so that $N_{\mathrm{par}} = N_\alpha$ (Cara et al., 2017). Scanning only the set of energies, elevations, and azimuths identified by the beam tracking strategy, requires

$\Delta t = N_E^* N_\theta^* N_\alpha^* \delta t / N_{\mathrm{par}}.$

The theoretical speed-up achieved by beam tracking then is

$$G = \frac{\Delta t_{\mathrm{full}}}{\Delta t} = \frac{N_E}{N_E^*} \frac{N_\theta}{N_\theta^*} \frac{N_\alpha}{N_\alpha^*},$$

corresponding to the fraction of VDF voxels that is sampled during each measurement cycle. Taking the THOR-CSW design as an example, a full energy–elevation–azimuth scan would have $N_E = 96$, $N_\theta = 32$, $N_\alpha = 32$. The standard energy tracking

mode has $N_E^* = 16$, $N_\theta^* = N_\theta$, $N_\alpha^* = N_\alpha$, so that $G = 6$. The standard combined energy and elevation tracking mode has $N_E^* = 16$, $N_\theta^* = 16$, $N_\alpha^* = N_\alpha$, so that $G = 12$, i.e. an order of magnitude improvement in time resolution can be achieved. In reality, the speed-up may be somewhat less since for angular beam tracking the importance of the settling times needed when changing the high voltages on the analyser plates is relatively higher (there are more frequent deflector voltage scans, while they are shorter). Note that the voltages on the deflector plates can be swept in a continuous manner, avoiding settling times except at

the start of an elevation scan, which coincides with the start of an energy step (Cara et al., 2017). For example, a VDF obtained from sampling all $N_E \times N_\theta \times N_\alpha = 98304$ voxels with an integration time of $\Delta t_{\mathrm{int}} = 0.180\,\mathrm{ms}$ and a high voltage settling time of $\Delta t_{\mathrm{hv}} = 0.200\,\mathrm{ms}$ would take $\Delta t_{\mathrm{full}} = N_E(N_\theta \Delta t_{\mathrm{int}} + \Delta t_{\mathrm{hv}}) = 573\,\mathrm{ms}$, given that all azimuths are acquired simultaneously. Energy tracking alone would sample $N_E^* \times N_\theta \times N_\alpha = 16384$ voxels in $\Delta t = N_E^*(N_\theta \Delta t_{\mathrm{int}} + \Delta t_{\mathrm{hv}}) = 95.4\,\mathrm{ms}$, exactly $G = 6$ times faster than a full scan. Combining energy and elevation tracking leads to sampling $N_E^* \times N_\theta^* \times N_\alpha^* = 8192$ voxels in only

$\Delta t = N_E^*(N_\theta^* \Delta t_{\mathrm{int}} + \Delta t_{\mathrm{hv}}) = 50\,\mathrm{ms}$. The resulting speed-up is $\Delta t_{\mathrm{full}}/\Delta t = 11.5$, slightly less than the expected $G = 12$.

## 3   Beam tracking strategies

The potential speed-up provided by beam tracking comes at a cost: There is a risk that one misses (part of) the solar wind beam. The reason is that one has to predict, at the start of a measurement cycle, where the beam is to be found. Such a prediction necessarily is prone to error. Therefore, one has to devise a beam tracking strategy that is robust.

### 3.1   Computing mean energy and arrival direction

As discussed above, beam tracking boils down to predicting the average velocity or energy of the solar wind beam, and its arrival direction. The energy, elevation, and azimuth sampling windows are then shifted so that they stay centred around the predicted value.

Let us consider internal beam tracking first. During VDF measurement cycle $p$, the instrument scans through a con-

tiguous subset of the energies $E_i$, $i = i_p, \ldots, i_p + N_{Ep}^* - 1$, of the elevations $\theta_j$, $j = j_p, \ldots, j_p + N_{\epsilon p}^* - 1$, and azimuths $\alpha_k$,




$k = k_p, \ldots, k_p + N^*_{\alpha p} - 1$, to obtain a distribution function $f(E_i, \theta_j, \alpha_k)$. Based on these measurements, one can determine the energy distribution by summing over the elevation and azimuth bins

$$f_E(E_i) = \sum_{j=j_p}^{j_p + N^*_{\theta p} - 1} \sum_{k=k_p}^{k_p + N^*_{\alpha p} - 1} \gamma_{ijk} f(E_i, \theta_j, \alpha_k),$$

where the $\gamma_{ijk}$ are known factors that incorporate instrument geometry, detector gain, and detector ageing coefficients. Note that the energy distribution can be constructed progressively as the scans over energy, elevation, and azimuth are performed. The mean or peak energy $\langle E \rangle_p$ can be readily derived from this energy spectrum; the former is considered to be a bit more robust than the latter. One can proceed in a completely analogous way to obtain the mean or peak elevation and azimuth.

The above description is actually a simplification that is applicable only to 3-axis stabilised or slowly rotating spacecraft. If the spacecraft spin phase changes significantly during the measurement, the construction of the VDF becomes more complicated as the attitude changes have to be accounted for; this is a task that usually is performed on-ground. Beam tracking, however, requires the mean energy and arrival directions to be established on-board and fast. First, one can simply assume that the spacecraft spin rate is sufficiently low. For the THOR-CSW case, the spin phase change should be less than $\Delta \omega = \arctan(1.5°/24°) = 3.6°$ during the acquisition of a VDF in order not to lose the desired angular resolution. Knowing that THOR was planned to spin at $2\,\mathrm{rpm}$, the VDF acquisition time should be less than $\sim 300\,\mathrm{ms}$. In practice, this condition may be somewhat too strict since most data are gathered near the centre of the sampled range. In any case, the faster a VDF is assembled, the less such rotational smearing effects; the use of beam tracking helps to ensure that this condition is satisfied. There is a simple way, however, to relax the above limitation. Rather than computing the energy distribution over the whole set of energies that have to be scanned, the set can be divided in a number of chunks, each of which covers only $N_{\mathrm{chunk}} \ll N^*_E \leq N_E$ energy channels. In the case of THOR-CSW, the choice $N_{\mathrm{chunk}} = 8$ was considered. A full energy scan would therefore require $N_E / N_{\mathrm{chunk}} = 12$ chunks, while a 16-energy scan requires 2 chunks. The (partial) moments are computed for each chunk and then combined to obtain the full moments while taking into account the spacecraft spin. Such an operation is much simpler than a full correction for spin at the level of the individual energy–elevation–azimuth voxels. It is convenient because the computations for each chunk can be done in parallel with the data acquisition for the next chunk. But most importantly, the condition for avoiding rotational smearing applies to the acquisition of a chunk, rather than of the full VDF. The time needed to collect a chunk with 8 energies and 16 elevations is $25\,\mathrm{ms}$, and that for a chunk with 8 energies and 32 elevations is $48\,\mathrm{ms}$, which both are well below the $\sim 300\,\mathrm{ms}$ limit found above. This offers a viable and straightforward way to compute the mean energy and arrival directions needed for internal beam tracking on-board. The same type of computation can provide all on-board plasma moments, which is particularly useful if only a fraction of all full VDFs can be transmitted to the ground due to telemetry limitations; this enables the implementation of a survey data mode that provides only the on-board moments with good quality.

External beam tracking is an interesting option when another instrument is available that provides plasma moments at a higher speed than the plasma spectrometer, such as a Faraday cup instrument (e.g. Šafránková et al., 2013). Such instruments can provide solar wind speed and velocity direction (and thermal velocity), from which the settings for the next measurement




cycle can be derived. Usually, the arrival direction is known in that instrument's reference frame. One then needs to know its alignment relative to that of the plasma spectrometer to be able to translate these measurements into usable values for the beam tracking procedure onboard. Finally, also the delay time between data acquisition and use in the plasma spectrometer must be known. The acquired data receive a time stamp from the clock of the auxiliary instrument, which must be synchronised to the

same reference as the plasma spectrometer's clock. The delay time includes computation time in the auxiliary instrument and the time needed for transmission, possibly via the payload processor; it obviously should be minimal.

### 3.2   Prediction

The decision on which part of phase space to scan in the upcoming measurement cycle is always a matter of prediction. The simplest form of prediction is just taking the value from the last measurement. For instance, if the previous cycle $p$ resulted in

an average energy $\langle E \rangle_p$, one can choose the center of the energy range for cycle $p + 1$ as

$$E^{(p+1)} = \langle E \rangle_p,$$

i.e. one uses zero order (constant) extrapolation. A slightly more advanced prediction is obtained through first order (linear) extrapolation:

$$E^{(p+1)} = 2\langle E \rangle_p - \langle E \rangle_{p-1}.$$

Second order (parabolic) extrapolation results in

$$E^{(p+1)} = 3\langle E \rangle_p - 3\langle E \rangle_{p-1} + \langle E \rangle_{p-2}.$$

In principle one may even use higher-order polynomial extrapolation. There are, however, a number of drawbacks. In general, $n$-th order extrapolation requires $n + 1$ preceding values. The underlying assumption of polynomial extrapolation is that the behaviour of $\langle E \rangle(t)$ is smooth ($n$ times continuously differentiable) during this whole $(n+1)\Delta t$ time period; if not, the extrap-

olated value may be completely off the mark. Such smoothness is questionable in the solar wind at shocks or discontinuities, so a high $n$ is not warranted. All in all, one can expect such techniques to work reasonably well only if the energy does not change rapidly, but in such cases a low order extrapolation works fine too.

Also, if any of these values happens to be corrupted (e.g. by a single event upset in one of the anodes or in the ADC electronics), the prediction can be wrong. In order to eliminate values that are completely off, a voting mechanism can be used.

Consider the three last measurements, and compute $|\langle E \rangle_p - \langle E \rangle_{p-1}|$, $|\langle E \rangle_p - \langle E \rangle_{p-2}|$, and $|\langle E \rangle_{p-1} - \langle E \rangle_{p-2}|$. Identify the smallest of these three differences. It can be assumed then that this smallest difference corresponds to two values that are not corrupted as they seem to agree with each other. One can then perform constant extrapolation by adopting the most recent of those two numbers as $E^{(p+1)}$. Alternatively, one can perform linear extrapolation with those two values. Note that such a more robust procedure requires one more preceding value, implying that the prediction may rely on information that is somewhat

older. In other words, the ability of the algorithm to cope with rapid changes in the solar wind VDFs is slightly degraded.



One way of implementing (internal or external) beam tracking is by storing the $\langle E \rangle_p$ measurements, together with their time tag, in a first-in first-out queue. As soon as the instrument is ready for setting up the next VDF acquisition, the most recent measurements are retrieved to make a prediction. This asynchronous system always works, even when there are processing delays associated with the interpretation of previously obtained VDFs (for internal beam tracking) or with the processing and

transmission of the data of the driving instrument (for external beam tracking). Such asynchronicity is also useful if the VDF acquisition cycle has a variable duration, e.g., when the number of sampled energy bins is variable.

The procedure outlined above also holds for angular tracking. There is one additional complication, though, in that all azimuth-elevation pairs must be rotated along with the spacecraft spin. Not doing so would lead to systematic offsets in predicted beam position, which can be neglected only for slowly rotating spacecraft.

An argument in favour of external beam tracking is that such an instrument may offer more recent data to base a prediction on. Nevertheless, it is important to note that, conceptually, internal beam tracking can always be considered "good enough". Indeed, a prediction based on the previous plasma spectrometer measurement involves an extrapolation over a time interval roughly equal to the VDF acquisition time. This would not be justified if the solar wind would change significantly over such an interval. But if that is the case, the time resolution of the spectrometer is simply insufficient and the VDFs that are acquired

are questionable anyhow since they involve sampling a changing distribution. A posteriori verification is always possible by comparing subsequent VDFs.

### 3.3   Beam loss detection and recovery

The desire for a robust prediction stems from the fact that the internal beam tracking process suffers from a self-destructive property: if a prediction is off the mark, the next measurement cycle will not correctly represent the VDF, so that the subsequent

prediction is extremely likely to be worthless. In other words, once one starts having difficulties with tracking the beam, one will rapidly miss it completely and possibly indefinitely.

One therefore needs a system for recovery of the beam. A straightforward and failsafe mode of operation is by regularly performing a scan over the entire energy–elevation–azimuth range. In this way, if one loses the beam, one is sure to pick it up again after a finite time interval. More sophisticated strategies could examine the shape of the obtained VDF to check whether

part of the VDF is missed. Implementing such sophisticated strategies on-board, however, is difficult and it is hard to ensure that they are robust (i.e. when there is beam loss, the strategy should indicate this) and efficient (i.e. when the strategy indicates that there is beam loss, that should actually be the case so that a beam recovery action is needed). In the present study we have adopted a simple condition: if the measured density is below a threshold $n_{\mathrm{beam-loss}} = 0.1\,\mathrm{cm}^{-3}$, the beam is considered to be lost. The recovery action is to scan over the entire instrument range once or several times, depending on the extrapolation

method, to restart the beam tracking process. In fact, this is exactly how the beam tracking strategy is initialised in the first place.

Beam loss is especially problematic if one is not able to downlink the full VDFs, but only moments that are computed on-board. In that case one has no means whatsoever to assess the reliability of the moments, since parts of the VDF might have been missed. It is then advised to downlink a subset of the VDFs, though at a much slower rate, to at least allow a regular check




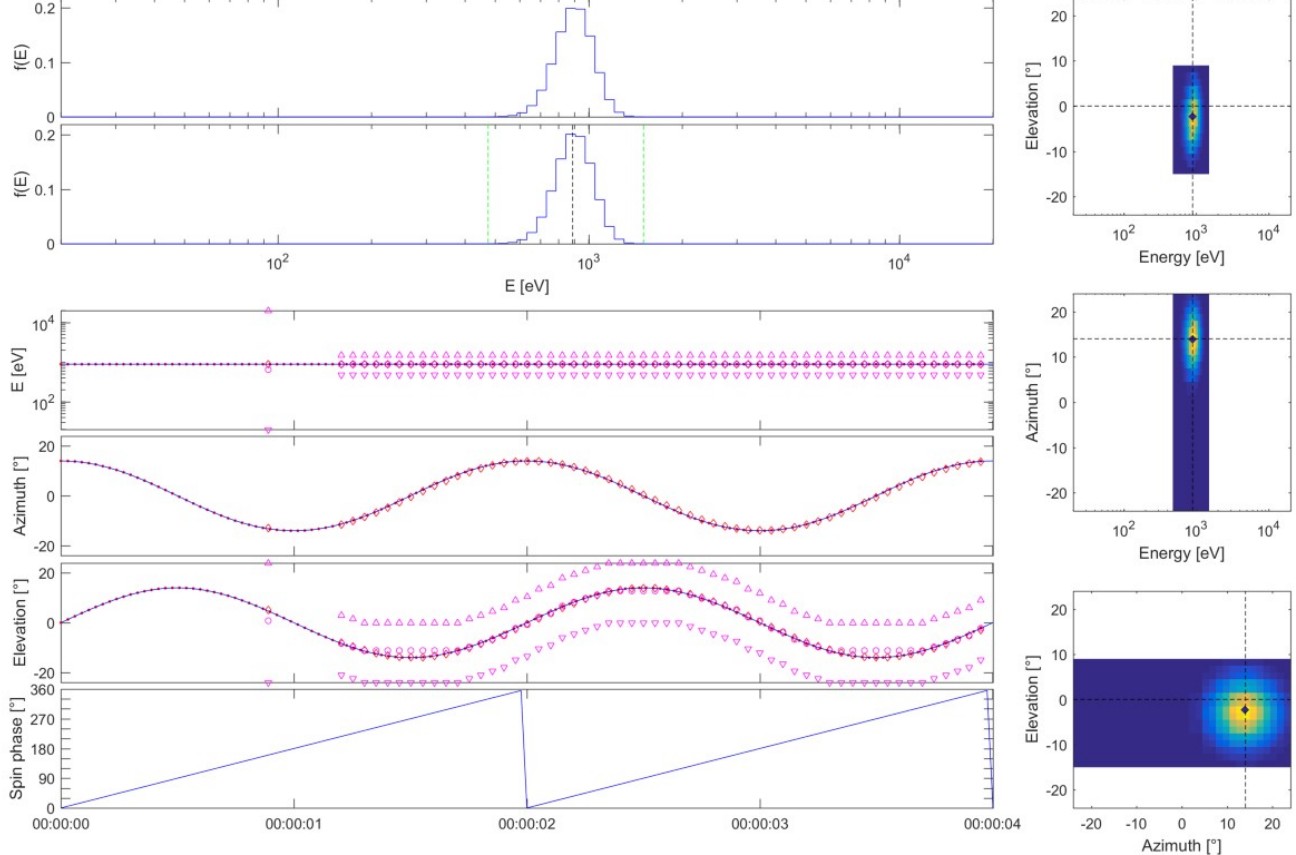

**Figure 1.** Plasma spectrometer measurements of a constant Maxwellian solar wind beam on a rapidly spinning spacecraft using internal energy and elevation beam tracking. From top to bottom: the energy spectrum of the Maxwellian solar wind; the energy spectrum as acquired by the plasma spectrometer at $t = 3.95\,\mathrm{s}$ with the vertical black and green dashed lines indicating the centre and the bounds of the sampled energy range; the energy as a function of time, where the horizontal blue line represents the true solar wind value, the small red dots are the Faraday cup measurements every $30\,\mathrm{ms}$ (not used with internal beam tracking), the magenta circles and triangles indicate the centre and the bounds of the sampled energy range, and the red diamonds give the mean energy as determined by the plasma spectrometer; the azimuth (same format, no beam tracking for azimuth); the elevation (same format); and the spin phase. The panels at the right hand side show the energy–elevation, energy–azimuth, and azimuth–elevation projections of the VDF at $t = 3.95\,\mathrm{s}$. See the main text for more details.

on the proper functioning of the beam tracking strategy. Alternatively, one may downlink reduced distributions, e.g. the energy and angular distributions $f_E(E_i)$, $f_\theta(\theta_j)$ and $f_\alpha(\alpha_k)$, to ascertain that no significant part of the population has been missed.



### 3.4 Physical underpinning

Losing the beam is definitely to be avoided if one aims for continuous and reliable solar wind measurements. The key question is: how rapid is the instrument VDF sampling compared to the variability in the solar wind?

An order-of-magnitude answer to this question is obtained by considering the following qualitative argument. Spatial vari-
ations in the ion distributions cannot be much smaller than the ion gyroradius, which is on the order of $100\,\mathrm{km}$ in the solar wind at $1\,\mathrm{au}$. A steady plasma discontinuity of such thickness that passes by the observer with a (fast) solar wind speed of $1000\,\mathrm{km}\cdot\mathrm{s}^{-1}$ and with the discontinuity normal aligned with the flow direction (the most pessimistic situation), is seen by the observer as a time variation over $100\,\mathrm{ms}$. In order to track abrupt changes at that time scale, a measurement time resolution of $\sim10\,\mathrm{ms}$ should be sufficient.

Another way to address this question is to look at some of the highest-cadence solar wind measurements ever made. Data from the Bright Monitor of Solar Wind (BMSW) experiment on the Spektr-R mission (Šafránková et al., 2013) indicate shock ramps that last only $200\,\mathrm{ms}$. A statistical analysis by Riazantseva et al. (2015) shows that the solar wind fluctuation spectrum becomes quite flat around $10\,\mathrm{Hz}$, indicating that rapid intermittent variations with rather large amplitude are fairly common.

One arrives at the conclusion that rapid variations do occur and that beam tracking works best for sampling frequencies of
$10$–$100\,\mathrm{Hz}$. If the plasma spectrometer succeeds in sampling the VDFs at such a high cadence, there is little risk for beam loss.

## 4    Performance

In this section different strategies for beam tracking are evaluated by means of a software simulator of the THOR-CSW instrument.

### 4.1    Beam tracking on a spinning spacecraft

As a first test, consider a constant solar wind proton beam in the form of an isotropic Maxwellian distribution, with a speed that does not coincide with the solar direction (i.e. with the spin axis of the spacecraft). We ignore here the issue of aberration. As the spacecraft spins, the beam appears to trace a circle around the spin axis in the spectrometer field of view. Angular beam tracking can then be used to follow this ever-changing apparent arrival direction. We consider a solar wind beam with a density of $5\,\mathrm{particles}\cdot\mathrm{cm}^{-3}$, a velocity of $[-400, 100, 0]\,\mathrm{km}\cdot\mathrm{s}^{-1}$ in GSE coordinates, an isotropic temperature of $10^5\,\mathrm{K}$, and
a spacecraft spin period of $2\,\mathrm{s}$. Internal energy and angular beam tracking are used with constant extrapolation.

Figure 1 shows the results of the simulation (see Supplementary Materials for animations of all the simulations presented in this paper). The instrument is initialised at time $t = 0\,\mathrm{ms}$. It starts measuring a first VDF over all energies and all angles at $t = 600\,\mathrm{ms}$, an operation that lasts almost $600\,\mathrm{ms}$. The mean energy, azimuth and elevation are determined; note that these measurements are associated with the middle of the time interval during which the VDF is acquired. The mean energy and
elevation then are used to start energy and elevation beam tracking. For the energy, the beam tracking procedure is useful at the beginning to find the appropriate energy range; as the beam energy remains constant, the energy sampling interval does



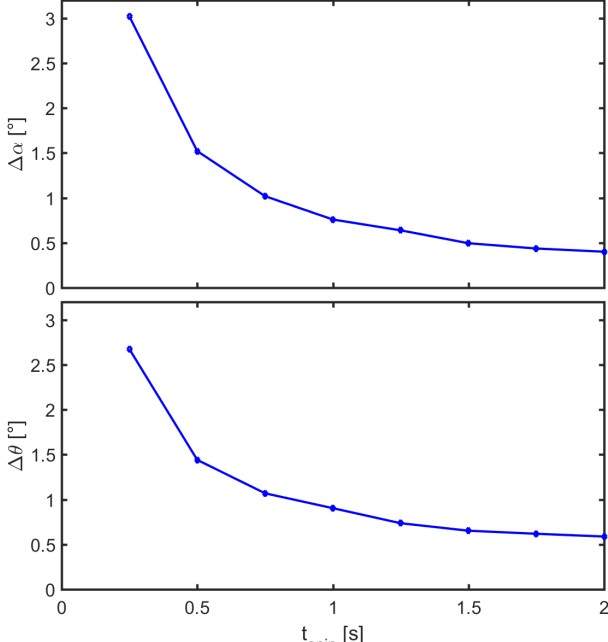

**Figure 2.** Plasma spectrometer measurements of a constant Maxwellian solar wind beam on a spinning spacecraft using internal energy and elevation beam tracking. The plot shows the maximum deviations $\Delta\alpha$ and $\Delta\theta$ between the spectrometer's mean azimuth and elevation and the solar wind azimuth and elevation as a function of the spacecraft spin period $t_{\mathrm{spin}}$.

not change any more. The elevation, however, changes sinusoidally. As can be seen in the figure, the beam is tracked very well, thanks to the prediction that takes the spacecraft rotation into account. Note that the centre of the sampled elevation range cannot follow the measured mean elevation when the upper or lower bound of the range coincide with the spectrometer's upper or lower elevation limit, but as long as the difference is small and the beam fits into the scanned range, there is no problem. The difference between the solar wind arrival angles and the measured mean azimuth and elevation remains below $0.6°$, well within the $1.5°$ discretisation error.

There is no risk for losing the beam in energy or elevation as its position in energy–elevation–azimuth space is constant when compensating for the spacecraft spin. It is interesting to see what happens if the spin rate changes. Variants of the above example have been simulated for $t_{\mathrm{spin}}$ from $0.25$ to $2\,\mathrm{s}$; for each of these, the maximum azimuth and elevation deviations have been evaluated over a full spin (while ignoring possible transient effects during the initialisation of the beam tracking mode). As Fig. 2 shows, the deviations become larger as the spacecraft spins faster. For example, with a spin period of only $0.25\,\mathrm{s}$ (see Fig. 3), the $50\,\mathrm{ms}$ time needed to collect a VDF is too large to justify the hypothesis that the solar wind does not change in the meantime (in the spacecraft frame of reference). Consequently, the collected distributions are somewhat distorted. Nevertheless, the beam tracking process still works fine.



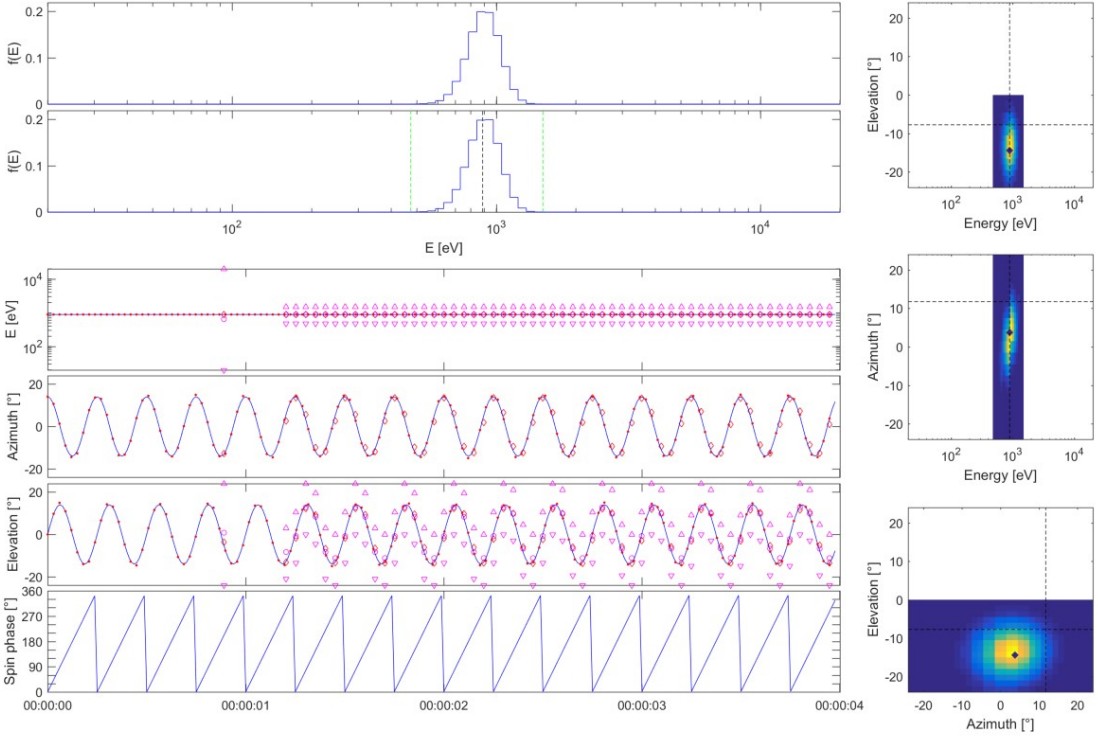

**Figure 3.** Plasma spectrometer measurements of a constant solar wind beam from a spacecraft with spin period $t_{\mathrm{spin}} = 0.25\,\mathrm{s}$. The plot layout is the same as that of Fig. 1.

## 4.2 Beam tracking at a plasma discontinuity

In a second test the response of the plasma spectrometer to the passage of a plasma discontinuity is examined. The discontinuity is characterised by a transition in proton properties as the density changes from 5 to $1\,\mathrm{particles\cdot cm^{-3}}$ and the isotropic temperature from $10^5$ to $4\times10^5\,\mathrm{K}$, while the velocity jumps from $[-400, -50, 0]$ to $[-800, 0, 100]\,\mathrm{km\cdot s^{-1}}$ in GSE coordinates. The transition is centred at $t = 2\,\mathrm{s}$ and has a duration $\Delta t_{\mathrm{disc}} = 500\,\mathrm{ms}$. The spacecraft spin period is $30\,\mathrm{s}$ but does not really matter here. Internal energy and angular beam tracking are used with constant extrapolation. The simulation in Fig. 4 demonstrates how both energy and angular beam tracking work in unison to flawlessly follow the solar wind beam as it changes its direction and as its energy increases by a factor of 4 through the transition. If one would have sampled over the full energy–elevation–azimuth ranges, there would have been only 1 or 2 measurements during the passage of the discontinuity, while there are ~10 measurements when using beam tracking.

The simulation in Fig. 5 repeats the previous example, but now for $\Delta t_{\mathrm{disc}} = 50\,\mathrm{ms}$. Given that the beam changes its energy considerably and abruptly, a situation of beam loss occurs during the transition. This is due to the energy change, not due to





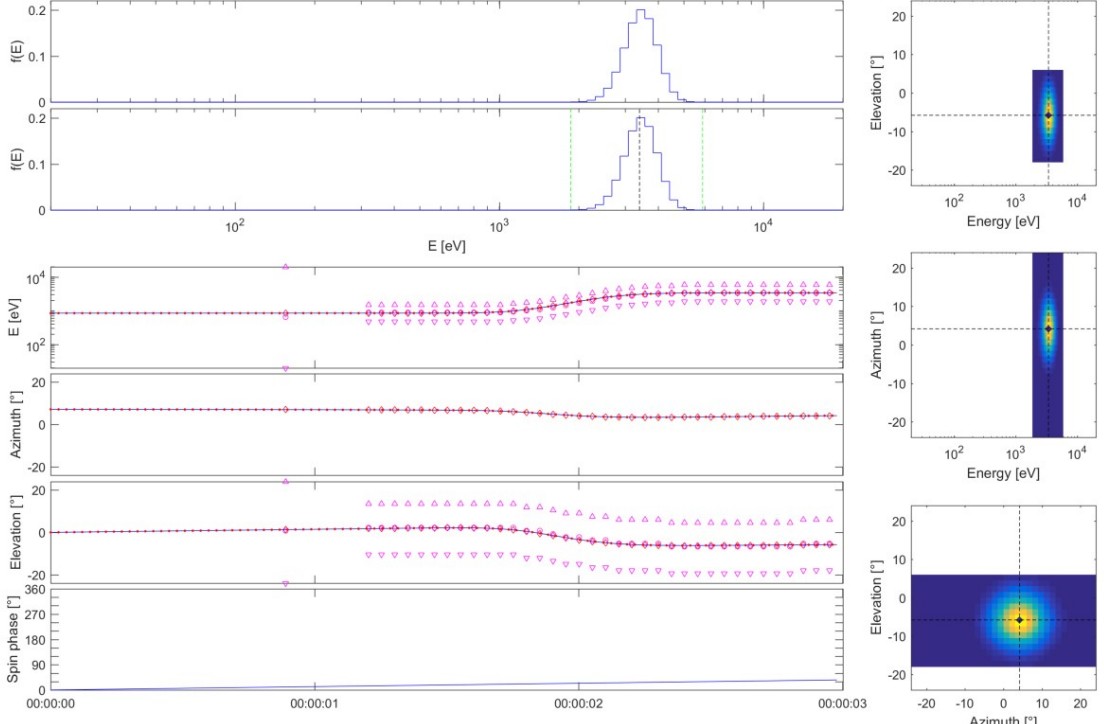

**Figure 4.** Plasma spectrometer measurements during the passage of a gradual plasma discontinuity (duration $500\,\mathrm{ms}$) using internal energy and elevation beam tracking. The plot layout is the same as that of Fig. 1.

the elevation change. The instrument has begun scanning over the lower energy channels at the time the solar wind velocity is ramping up rapidly, so that the solar wind beam has disappeared from the higher energy channels in the scan. This leads to an underestimation of the density, and to a decrease of the mean energy so that the next VDF measurement cycle is completely off. Missing the beam leads to a measured density that is less than the $0.1\,\mathrm{particles}\cdot\mathrm{cm}^{-3}$ threshold, triggering the beam loss

5 condition at the end of acquiring the data point at 00:00:02.050 (collection between 00:00:02.025 and 00:00:02.075). The figure shows the beam recovery strategy jumping into action by first doing a full scan to find the beam again at 00:00:02.365 (data collected between 00:00:02.075 and 00:00:02.655) and then restarting beam tracking to resume high cadence data production (first data point at 00:00:02.680 collected between 00:00:02.655 and 00:00:02.705).

In order to explore the limits of beam tracking as the discontinuity time scale becomes shorter, the maximum density and

10 energy errors (deviation of the measured moments from the solar wind value) are evaluated as a function of $\Delta t_{\mathrm{disc}}$ and are presented in Fig. 6. The top panel in the figure indicates whether or not beam loss occurs (true or false, respectively). When there is beam loss, the density is erroneous by definition since it is below the threshold there. Note that the error may already be important even when the beam loss condition is not triggered yet. The energy, azimuth and elevation errors also systematically





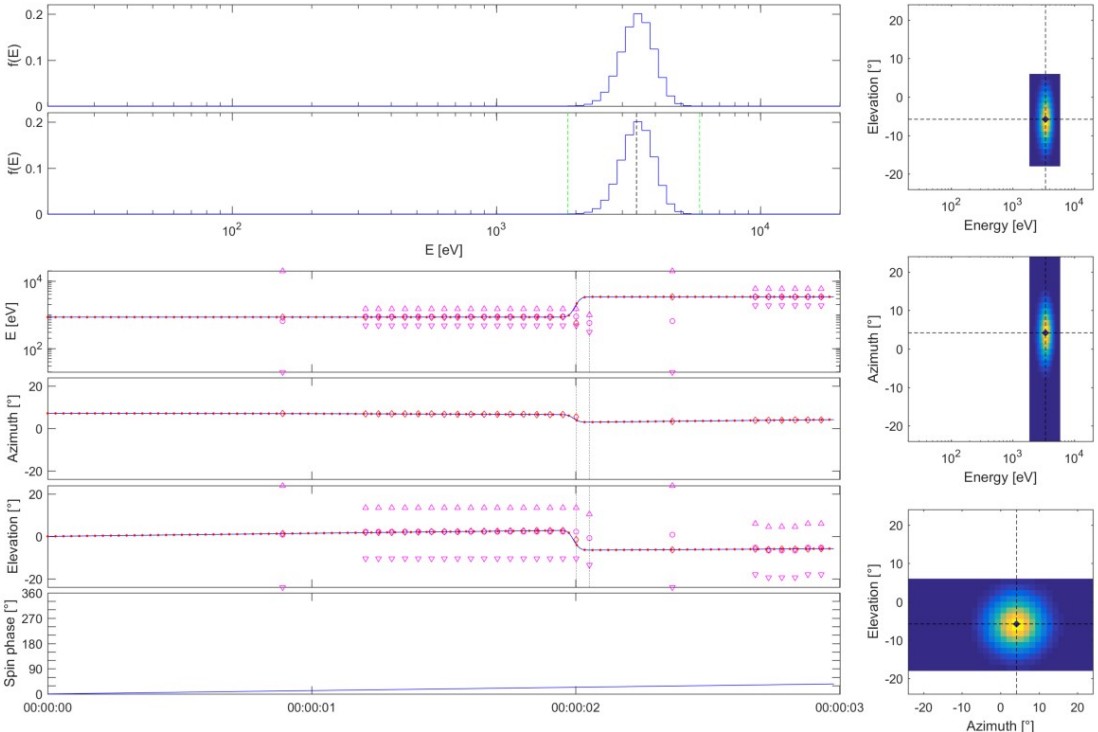

**Figure 5.** Plasma spectrometer measurements during the passage of an abrupt plasma discontinuity (duration $50\,\mathrm{ms}$) using internal energy and elevation beam tracking. The plot layout is the same as that of Fig. 1.

increase for a more rapid transition. While the maximum azimuth and elevation errors remain $\leq 0.75°$ (half of the $1.5°$ the angular resolution) as long as there is no beam loss, the maximum energy deviation is around $100\,\%$, which is not surprising since the beam is lost because it moves out of the energy range. The measurement points right before beam loss can thus be erroneous as part of the distribution may already be missed. One might fit an analytical distribution function (Maxwellian, bi-Maxwellian, Lorentzian) to the observed VDF to try to compensate for that. In any case, a look at the VDF will help in identifying that there has been an issue and to ascertain that a part of the VDF has not been measured.

In conclusion: Beam tracking can deal with progressive changes over a time scale longer than the sampling time, regardless the magnitude of the change. For shorter time-scale changes, there is no problem as long as the changes are not very large so that the beam still fits in the energy and angular windows.

## 4.3 Beam tracking for fast solar wind measurements

In the previous examples, synthetic data have been used to understand the possibilities and limitations of beam tracking. We now try to perform more realistic tests. Since no full solar wind VDF measurements have ever been made at such a rapid





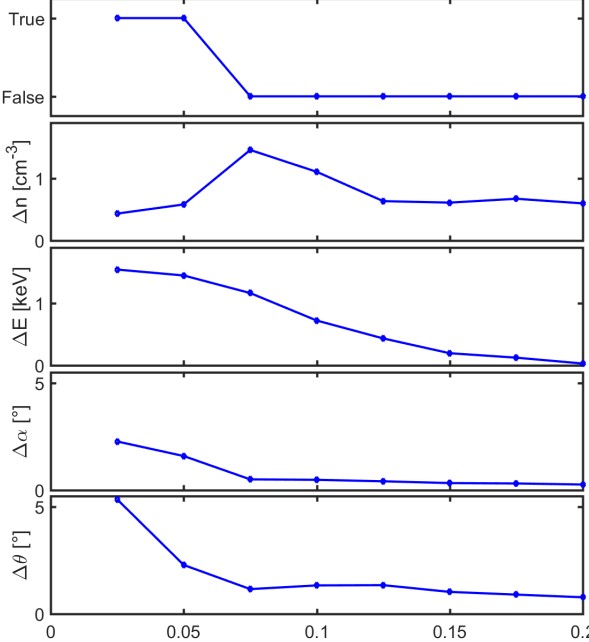

**Figure 6.** Plasma spectrometer measurements during the passage of a plasma discontinuity. The spectrometer uses internal energy and elevation beam tracking. The plot shows the occurrence of beam loss (true or false) and the maximum deviations in plasma density, energy, azimuth, and elevation between the measured values and the true solar wind values that occur throughout the passage, as a function of the discontinuity crossing duration $t_{\mathrm{disc}}$. The measurements are more accurate as the plasma property changes associated with the discontinuity occur over a longer time scale.

cadence, we have to create hypothetical solar wind data. This is done by using the aforementioned high-cadence solar wind measurements from the BMSW experiment on the Spektr-R mission (Šafránková et al., 2008, 2013). The moments from that instrument, with a time resolution of $\sim$31 ms, have been used to construct Maxwellian proton distributions, and the resulting VDF time sequence has been used as the "true solar wind" sampled by the plasma spectrometer. A simulation is shown in

5  Fig. 7 for BMSW measurements on 2014-06-08 exhibiting moderate changes in solar wind direction; there is little variation in density, energy, and some variability in temperature. The instrument is perfectly capable of following these changes since these are neither dramatic in magnitude nor very abrupt as they occur over time scales of seconds. Indeed, there do not seem to be discontinuous variations in the BMSW data, implying that solar wind variability takes place mostly over time scales of a multiple of $\sim$31 ms.

10  A more challenging situation is presented in Fig. 8. The BMSW instrument observes a strong shock around 2015-06-22 18:28:22 UT, where the velocity changes from $400$ to $700\,\mathrm{km \cdot s^{-1}}$, accompanied by solar wind direction changes, and by density and temperature enhancements by a factor of 2 to 3. The variations are both large and fast. The Faraday cup measurements at this time were performed using sub-optimal high-voltage settings that lead to an overestimation of velocity and temperature and





**Figure 7.** Plasma spectrometer measurements for a solar wind simulation based on BMSW on Spektr-R observations on 2014-06-08, using internal energy and elevation beam tracking. The plot layout is the same as that of Fig. 1, but also shows density, GSE velocity in the spacecraft frame of reference (including spacecraft spin, the spacecraft's orbital motion around Earth, and the Earth's motion around the Sun) and temperature, as a function of time.

an underestimation of density; the velocity overshoot up to $900 \, \mathrm{km \cdot s^{-1}}$ is likely unphysical. In the present exercise we ignore these data reliability issues and blindly feed the simulation with the Faraday cup moments. It turns out that the beam tracking procedure works perfectly. While the solar wind energy changes significantly in about 2 seconds, this change occurs stepwise




and with the instrument's $50\,\mathrm{ms}$ time resolution there are sufficient intermediate samples to follow the energy enhancement. The beam direction shows rapid changes between 18:28:18 and 18:28:22 UT and between 18:28:33 and 18:28:38 UT, and these too are well tracked. Although beam tracking works well, several problems are apparent. First, the $[-24°, +24°]$ elevation and azimuth ranges are sometimes too small. For instance, the solar wind beam elevation reaches $+20°$ around 18:28:13 UT, which

is too close to the limits of the instrument, and the solar wind azimuth effectively goes beyond the $-24°$ limit around 18:28:21 UT. In both cases, part of the solar wind beam is missed. Especially for the latter this leads to an error on the measurement, especially for the density. Such strong angular deviations are rare, but that makes these situations particularly interesting from the scientific point of view. Note also that situations in which the beam leaves the instrument field of view will occur even more often if there is a deviation of the pointing of the instrument (i.e. of the spacecraft it is mounted on) from the solar direction.

A second issue is that during the most rapid parts of the transitions around 18:28:22, the solar wind distribution changes too abruptly so that the VDF is mixed up (especially apparent in the animated version of the simulation in the Supplementary Materials). This situation is at the limits of the transition time scale inferred in section 3.4: the magnetic field can be strong near interplanetary shocks, and so the gyroradius might be relatively small; combined with a large speed, this can lead to short time scales. A third problem is that around 18:28:22.5 the solar wind temperature is at moments so high that the beam

becomes too broad to be captured completely in the sampling window; the density and the temperature as determined by the instrument are therefore somewhat too small. Sampling the solar wind without beam tracking every $600\,\mathrm{ms}$ partially avoids the high temperature issue, but the assumption that the VDF does not change during the sampling interval would be justified even less. All solar wind measurements up to now have had to contend with that. The speed-up from beam tracking appears to be essential to overcome this difficulty.

## 4.4 Internal and external beam tracking

The $50\,\mathrm{ms}$ time resolution of the plasma instrument with energy and elevation tracking described above is of the same order as that of a typical Faraday cup instrument. In that situation, there is little to be gained by using external rather than internal beam tracking. If one decides to run the plasma instrument using energy tracking only (16 energies, 32 elevations), for instance, in order to keep a field of view that is as wide as possible, the time resolution is $\sim 100\,\mathrm{ms}$, i.e., significantly slower, and then

external beam tracking becomes attractive. This situation is shown in Fig. 9 for an assumed delay time (time between centre of Faraday cup measurement and the moment that it is available for the plasma spectrometer) $\Delta t_{\mathrm{delay}} = 30\,\mathrm{ms}$. The error on the Faraday cup measurements should be on the order of the spectrometer energy and angular resolution at most. The hypothesis made here is that they are exact. Again, beam tracking works well, but the risk of time variability below the VDF acquisition time scale is even larger than before. This illustrates the fundamental limitation of external beam tracking. Fast VDF acquisition

is needed both to avoid variability while acquiring a VDF, and to have a reliable prediction for beam tracking thanks to a short prediction horizon. External beam tracking only addresses the second issue. An advantage of external beam tracking is that beam loss cannot occur and a recovery strategy is not needed: If the instrument keeps following the guidance from the Faraday cups (and assuming that these produce accurate results), it will always recover the beam, even if the beam has disappeared from the instrument field of view for some time.





**Figure 8.** Plasma spectrometer measurements for a solar wind simulation based on BMSW on Spektr-R observations of a strong shock on 2015-06-22, using internal energy and elevation beam tracking. The plot layout is the same as that of Fig. 7.

## 5 Conclusions

Beam tracking is an important element in the observational strategy of plasma spectrometers that try to provide high-cadence solar wind ion VDFs for in-depth studies of the behaviour of the plasma and its response to turbulence at kinetic scales. It is an essential tool to guarantee optimal energy and angular resolution, without compromising the signal-to-noise ratio, with





**Figure 9.** Plasma spectrometer measurements for a solar wind simulation with the same data as Fig. 8, using external energy beam tracking with a delay of 30 ms. The plot layout is the same.

minimal VDF acquisition time. It requires the VDF acquisition rate to be fast enough so that the beam energy and direction do not change dramatically within the acquisition time interval. At the same time, trustworthy run-time predictions of beam energy and direction must be available, either from the previous measurements (internal beam tracking) or from another instrument (external beam tracking). We have explored the performance of various beam tracking strategies using synthetic and actual



data from the Spektr-R/BMSW instrument. It turns out that the approach works well, but may fail at times, so that a robust beam recovery mechanism must be planned (for the case of internal beam tracking).

It appears that solar wind variations can at times be extremely rapid, as for the interplanetary shock observed on 2015-06-22 around 18:20:22 UT by the Spektr-R/BMSW instrument, therefore requiring a high time resolution. The simulation experiments conducted here show that a time resolution of $50\,\text{ms}$ is sufficient for most situations, but at some fast shocks this is apparently not fast enough. In view of considerations regarding the proton gyroradius, is seems likely that a resolution of $\sim 10\,\text{ms}$ would be sufficient, but at present data at a $100\,\text{Hz}$ cadence are not available to verify this.

It is always advised to perform regular diagnostics to check whether the beam tracking strategy is working properly. This can be done by examining the VDFs that are recorded, from which it may be apparent that part of the solar wind beam is missing. It is therefore desirable to have a Faraday cup instrument and a plasma spectrometer working in tandem. Even though the usefulness of external beam tracking is limited, the Faraday cup measurements can be used for cross-calibration, to verify whether the beam does not move out of the field of view (partially or completely) and to assess whether beam loss has occurred (especially in situations where only the plasma spectrometer moments are available), and to verify whether the plasma distribution did not dramatically change while the spectrometer was acquiring a VDF.

Beam tracking is not to be confounded with a posteriori peak tracing as used on the Helios-1 and -2 spacecraft (Rosenbauer et al., 1977, 1981). Peak tracing consists in searching for the main peak position in an acquired VDF, which typically contains many voxels with little or no counts in case no beam tracking is used. One may then choose to retain only that part of the distribution function for downlink. Even if one does not perform such a peak search, modern data compression techniques are able to exploit the presence of empty bins to reduce the data volume efficiently. Beam tracking itself already provides such a data compression simply by not measuring irrelevant regions of energy–elevation–azimuth space.

An outcome of the simulations presented here is that a field of view of $48°\times48°$ (as originally foreseen for THOR-CSW (Cara et al., 2017)) tends to be a bit narrow. Enlarging the field of view would lead to a degradation of angular resolution (for the same number of azimuth and elevation bins), but a $2°$ angular resolution and a $64°\times64°$ field of view could be an interesting choice that simultaneously mitigates the problem of partially missing the beam when the solar wind velocity is strongly non-radial, deals with hot solar wind situations, and reduces the risk of beam loss when the solar wind arrival direction changes rapidly. Such a wider field of view also relaxes the constraint that the instrument should be pointing accurately to the average (aberrated) solar wind direction; allowing the pointing direction to be off by several degrees reduces the frequency of spacecraft attitude change manoeuvres. The downside is that deflection over large angles is difficult to achieve while respecting the desired angular resolution.

While beam tracking is extremely well suited for solar wind monitoring, it can be used in other contexts as well. A possible application would be to apply energy and angular beam tracking for focusing on the details of precipitating and upwelling ion or electron beams in the auroral regions: such beams typically are narrow in angular extent as they tend to follow the magnetic field, and they are nearly mono-energetic with an energy that can range from a tens of eV up to $\sim 10\,\text{keV}$, at least for electrostatically accelerated particles.



*Data availability.* Movies (in MP4 format) that illustrate the beam tracking simulations described in this paper are provided as supplementary material under doi:10.18758/71021039 at *http://repository.aeronomie.be/?doi=10.18758/71021039*. The Spektr-R/BMSW high resolution solar wind data can be obtained from *https://aurora.troja.mff.cuni.cz/spektr-r/*.

*Competing interests.* JDK is Topical Editor of Annales Geophysicae. None of the other authors has competing interests.

5   *Acknowledgements.* JDK, EN, SB, and BB at BIRA-IASB acknowledge support by the Belgian Science Policy Office through Prodex/THOR-CSW-DEV (PEA 4000116805). Work by BL and AF at IRAP was supported by CNRS and CNES. LP was supported by the Czech Grant Agency under Contract 16-04956S. MFM and DB acknowledge support by the Agenzia Spaziale Italiana under Contract No. ASI-INAF 2015-039-R.O. The authors gratefully acknowledge the scientific, technical, and managerial guidance by A. Vaivads and A. Wielders during the THOR Phase A study.



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
