# Peer review of "Beam tracking strategies for fast acquisition of solar wind velocity distribution functions with high energy and angular resolutions"

_Annales Geophysicae, 2018_

## Referee Comment (RC1) · Anonymous Referee #1 · 20 Jun 2018

Manuscript ID:  angeo-2018-59
Author:  De Keyser et al., 2018

Summary:
================================================================================
The manuscript describes a new instrument concept for measuring the solar wind core ion beam
using a plasma spectrometer mounted on a sun-pointed spinning spacecraft.

Despite my somewhat lengthy set of comments below, I greatly enjoyed reading the paper and
would like to see the realization of this technology come to fruition.  Most of my
concerns/comments below are either minor or general musings and some do not really require the
authors to take action or respond (I noted these where necessary).  Thus, I think the paper is
suitable for publication in Ann. Geophys.

================================================================================
General Comments:
================================================================================
--  Table 1 and Section 2:
   --  I would check to make sure that the shock jumps are correct, as I recall from the CfA
   Shock Database that several shocks had $\Delta V > 200$ km/s.
   https://www.cfa.harvard.edu/shocks/wi_data/
   --  You should reference some recent work that provides the first long-term statistical study
   on solar wind parameters near 1 AU by Wilson et al. [2018] (Note the supplemental material
   does separate parameters by fast and slow wind).
   --  I doubt either of these will modify the values in your table very much, but they will
   provide at least a reference/source for the provided values.
--  Section 2.1:  [The following are my musings, but are most likely not critical]
   --  I see you addressed most of my concerns below in Section 3 already, but I leave it here
   for reference.
   --  One thing of which to be careful are secondary/reflected ions near strong collisionless
   shocks.  I assume you have thought of this and know how to handle it, but I should mention
   that even when the reflected to incident ion density is relatively low, it can affect the
   bulk flow velocity estimate determined from typical velocity moment software significantly.
   If the spacecraft on which the instrument of interest in this paper is to orbit Earth and
   not, say, L1, then bow shock reflected ions will be an issue and the fraction of
   reflected-to-incident is much higher (>25% in some cases) than typical interplanetary shocks.
   This can affect the bulk flow velocity causing it to devaiate away from the core solar wind
   proton beam by upwards of 30%, i.e., >100 km/s [e.g., Wilson et al., 2014a].  In the case of
   a sun-pointed spinner on an outbound pass, the number of reflected ions entering the detector
   will likely be small, so probably not an issue.  However, the reflected ions at earthward
   propagating interplanetary shocks will always be an issue.  The primary difference is that
   most interplanetary shocks do not reflect a significant enough fraction of the upstream ions
   to generate much of a foreshock, so perhaps this is not cause for concern?
   --  I know of at least one interplanetary shock that caused problems for the PESA Low
   detector from Wind/3DP that was seen on 2001-11-24 near 05:51 UT.  The thermal energies got
   so large that the instrument lost the solar wind beam and did not enter tracking mode because
   it thought it was still following the beam.  Granted, the mode was not as well designed as
   newer spacecraft that use NV (i.e., roughly the count rate) but it is worth considering.
--  Section 3.4:
   --  Be careful with the estimates of the spatial scales for discontinuities.  The thickness
   of the shock ramp is not on ion scales, but on electron scales [e.g., Hobara et al., 2010;
   Mazelle et al., 2010].  What is not shown in the Spektr-R data is what was assumed for years

to be the actual shock ramp but was undersampled [e.g., see Wilson et al., 2012, 2017].  In general, I think your estimates are fine, but the statement that ion properties cannot change faster than ion scales is factually incorrect.  Further, it is not the case that the fluctuations discussed in the above references have no effect on the ions, as shown by Goncharov et al. [2014].
-- Section 4.1
  -- I am confused.  If you have a sun-pointed spinning spacecraft and you align the central elevation angle bin with roughly the Earth-sun line, why does the solar wind beam vary with spin in the elevation angle?  Or am I misunderstanding Figure 1 and the discussion in this section?  Is the spacecraft spin axis not aligned with the Earth-sun line?
  -- Page  7, Lines 27-30:  I do not follow the sentence starting with "The difference between..."  Is this a comment on the results shown in Figure 1 or a general comment about the solar wind?
  -- Page  8, Lines 4-5:  Can you be a little more quantitative with the statement "...distributions are somewhat distorted..."?  Distorted in what way?  Would one interpret the VDFs as having a higher temperature than reality, for instance?  If so, by how much?
-- Section 4.3
  -- Having had several long conversations with Drs. Safrankova and Nemecek (a few years ago now) about the capabilities and limitations of the BMSW instrument, I am curious how you managed to get the data into GSE coordinates.  It was my understanding that there is no way to know the actual spacecraft orientation and attitude necessary to rotate the data out of spacecraft coordinates into a physically meaningful basis.  Has this issue been recently resolved?
  -- The shock on 2015-06-22 arrived at L1 at ~18:08:24 UT (e.g., I looked at Wind data on CDAWeb).  Regardless, the bulk flow velocity along X-GSE jumps to nearly -800 km/s in the downstream and the ion temperature exceeds 100 eV (i.e., ~1.2 MK), so the temperatures may not be too inaccurate from BMSW.  The CfA shock database shows a density compression ratio of ~3.4 but I think the temperature changes by a factor >4-5.  [These are just comments, not really actionable items.]
  -- Page  9, Lines 50-51:  Are the temperature and temperature anisotropy significantly affected as well, or just the density moment?

-- Hot and/or Tenuous VDFs
  -- One of the biggest issues that I did not see addressed in the manuscript occurs during intervals when the density is low [i.e., below ~1 cm^(-3)] or the temperature is high (i.e., Ti > ~100-200 eV, depending on the instrument).  If we assume a bi-Maxwellian or even an isotropic Maxwellian, the peak phase space density goes as N*T^(-3/2).  The one-count level during the same interval does not drop/change relative to an adjacent, earlier interval.  Thus, the signal-to-noise ratio can drop preciptously during these periods.  I realize this is an issue faced by all particle instruments, but it is worth discussing to ensure you do not lose the critical parts of the distribution downstream of strong shocks with high temperatures but relatively low density (e.g., for really low upstream density).

===============================================================================
Minor Concerns:
===============================================================================
-- Page  1, Lines 35-50:  You could also mention waves and instabilities [e.g., Malaspina et al., 2013], as electromagnetic fluctuations are not solely limited to turbulence.  It is also important to measure the full 3D VDFs for analysis of instabilities.
-- Page  2, Lines 2-18:  The Wind spacecraft's 3DP instrument suite is also relevant here [e.g., Lin et al., 1995].

-- Page  2, Line 47:  I know voxel is a term analogous to a velocity-space pixel, but could you provide a definition for the reader that may not know this.
-- Page  7, Lines 10-12:  I am not sure I understand the sentence starting with "It starts measuring..."  You state the instrument starts sampling at 600 ms and the duration required to obtain one full VDF is another 600 ms.  Is that correct?

================================================================================
Typos, Grammar, etc.:
================================================================================
[The following are suggestions, not requirements (e.g., I do not recall rules for British vs. American grammar for when to use commas after things like "e.g." or "i.e.")]
Page  4, Line 25:  "12, i.e. an order"  -->  "12, i.e., an order"
Page  5, Line 56:  "i.e. one uses"  -->  "i.e., one uses"
Page  5, Lines 77-79:  "In order to eliminate values that are completely off, a voting"  --> "In order to eliminate outliers, a voting"
Page  5, Line 87:  Try rephrasing the following "Note that such a more robust procedure requires" as it is awkwardly phrased and not clear what is meant.
Page  6, Line 38:  "robust (i.e. when"  -->  "robust (i.e., when"
Page  6, Line 40:  "...cient (i.e. when"  -->  "...cient (i.e., when"
Page  6, Line 98:  "direction (i.e. with"  -->  "direction (i.e., with"
Page  8, Lines 62-63:  "The measurement points"  -->  "The measurements"
Page  9, Line 19:  "neither dramatic in magnitude nor very"  -->  "neither dramatic in magnitude or very"
Page 11, Line  5:  "instrument (i.e. of"  -->  "instrument (i.e., of"
Page 14, Line  5:  "manoeuvres"  -->  "maneuvers"

================================================================================
References:
================================================================================

-- Goncharov, O., et al., "Upstream and downstream wave packets associated with low-Mach number interplanetary shocks," Geophys. Res. Lett. 41, pp. 8100--8106, doi:10.1002/2014GL062149, 2014.
-- Hobara, Y., et al., "Statistical study of the quasi-perpendicular shock ramp widths," J. Geophys. Res. Vol. 115, pp. A11106, doi:10.1029/2010JA015659, 2010.
-- Lin, R.P., et al., "A Three-Dimensional Plasma and Energetic Particle Investigation for the Wind Spacecraft," Space Sci. Rev. Vol. 71(1), pp. 125--153, doi:10.1007/BF00751328, 1995.
-- Malaspina, D.M., et al., "Electrostatic Solitary Waves in the Solar Wind: Evidence for Instability at Solar Wind Current Sheets," J. Geophys. Res. Vol. 118, pp. 591--599, doi:10.1002/jgra.50102, 2013.
-- Mazelle, C., et al., "Self-Reformation of the Quasi-Perpendicular Shock: CLUSTER Observations," Proc. 12th Int. Solar Wind Conf., AIP Conf. Proc. 1216, pp. 471--474, doi:10.1063/1.3395905, 2010.
-- Wilson III, L.B., et al., "Observations of electromagnetic whistler precursors at supercritical interplanetary shocks," Geophys. Res. Lett. Vol. 39, L08109, doi:10.1029/2012GL051581, 2012.
-- Wilson III, L.B., et al., "Quantified energy dissipation rates in the terrestrial bow shock: 1. Analysis techniques and methodology," J. Geophys. Res. Vol. 119, pp. 6455--6474, doi:10.1002/2014JA019929, 2014a.
-- Wilson III, L.B., et al., "Revisiting the structure of low-Mach number, low-beta, quasi-perpendicular shocks," J. Geophys. Res. Vol. 122, pp. 9115--9133,

doi:10.1002/2017JA024352, 2017.

--  Wilson III, L.B., et al., "The Statistical Properties of Solar Wind Temperature Parameters Near 1 au," Astrophys. J. Suppl. Vol. 236(2), pp. 41, doi:10.3847/1538-4365/aab71c, 2018.

---

## Referee Comment (RC2) · Anonymous Referee #2 · 23 Jul 2018

General Comments:

The presented manuscript presents and discusses a novel approach to employ electrostatic spacecraft analyzers fitted with angular deflectors. By evaluating beam parameters of the surrounding plasma, only energy- and directional bins relevant to resolving said beam need to be sampled, resulting in much faster signal acquisition and as a result, higher time resolution.

The presented method represents an instance of a sparse sampling approach, in which the sample points from a high-dimensional parameters space are deliberately constrained to certain subsets of that space in order to obtain a maximum amount of

information with minimal sampling requirements. Similar techniques have been employed with great success in Biophysics (Such as compressed sensing techniques in neurosciences [1]), Astronomy (in aperture synthesis for telescopes [2]). Likewise in the same field as this manuscript, kinetic simulation approaches in space physics employ similar techniques to reduce the computational load of high-dimensional simulation spaces [3,4].

References to similar approaches from those fields, as well as overview papers of compressed sensing methods should be added, since a large body of general theoretical background work from other fields can be applied for this approach.

Specifically, the presented manuscript discusses a method to sparsely sample space plasma velocity distributions, with the intention of tracking a "beam" and sampling it with a minimum number of required samples, to obtain an extraordinarily high temporal resolution.

The model assumptions going into the example analysis performed in this manuscript are a) that the "interesting" part of the particle distribution is quite compact in shape, more precisely, in this analysis it is assumed to be maxwellian b) that it's overall shape stays the same, and only it's parameters change. These assumptions preclude the possibility of multiple mixed plasma distributions, such as a core and beam setup in a foreshock, rings or loss cones in a fermi-type acceleration region or any other non thermally-relaxed particle distribution.

I assume that the authors only focus on the solar wind distributions' core is motivated by their specific research interests. However, the study of kinetic physics of the solar wind, including the effects of turbulence, shocks and magnetic reconnection depends strongly on the ability to study and understand nonthermal distribution functions, that is, precisely those distribution functions that do not fulfill the assumptions going into the manuscript at hand.

While a much more thorough analysis and quantification of the detector behaviour

for realistic distribution functions will be required before the presented method can be employed in an actual instrument, it is probably not within the scope of this paper to perform them – the central subject and conclusion being the presentation and motivation of a sparse sampling scheme in the first place. Still, some more reflection on the limitations of the presented analysis, and avenues to further refine the analysis should be included.

In conclusion, this manuscript presents a thoroughly novel idea that merits publication and discussion in the wider scientific community, but suffers from being too narrow in it's goals and scope. After some major revisions, in which the presented method is evaluated with a focus on more general kinetic-physics processes, I consider it suitable for publication.

Specific Comments:

The prediction method presented in section 3.2 and it's discussion of polynomial extrapolation overshoots is very similar in nature to the problem of flux limiters in finite volume simulation methods, such as MHD simulation. There, too, the extrapolation of a reconstruction polynomial is clamped to remain within physically realistic boundaries. This similarity could be discussed and referenced (such as [5]).

The same section claims that "All in all, one can expect such techniques to work reasonably well only if the energy does not change rapidly", and I agree with that statement. However, especially in shocks, discontinuities and reconnection regions, where this assumption does not hold true, is where the most interesting kinetic plasma physics effects occur.

Note that the sudden changes of distribution function in these events are not simply a parameter change of a maxwellian: the shape of the distribution function departs *significantly* from a maxwellian whenever kinetic physics comes into play. If the spacecraft changes it's magnetic connection to a shock, beam distributions of highly nonthermal shape can suddenly "appear" outside of the thermal velocity radius of the

previous maxwellian. In reconnection regions, spitzer orbits and crescent-shaped velocity distributions additionally appear on top of any thermal background that might still be present. Additionally, nonisotropic superthermal tails can deform the solar wind distribution away from a maxwell-boltzmann shape.

The discussion in section 4.4, comparing internal and external beam tracking, is thus incomplete, as the asumption of a continuous change of maxwell distribution parameters won't represent reality in many interesting kinetic physics scenarios.

As for the beam loss criterion itself (sections 3.3 and 3.4), it is based on the assumption that the "beam" encompasses the entire interesting part of the distribution function at time of tracking, and that the only noteworthy change at a plasma discontinuity would be a sudden loss of the beam at one spot, with reappearance at another. This is a rough oversimplification of the wide variety of foreshock distribution functions (compare [6]): in many cases, additional beam distributions will occur far outside the thermal velocity extents of the solar wind beam, thus remaining untracked by the restricted sampling process presented here. "Beam Loss" as defined in this paper is neither an appropriate, nor a sufficient criterion for re-scanning of the complete velocity space.

The presented tests inadequately asses the response of the method to these kind of scenarios. While it is good and correct to assess the ability of this method to re-acquire the beam after a beam loss scenario with realistic dynamic timescales, this is, by far, not the only relevant measurement quantity to optimize for. I would suggest expanding section 4 with a discussion of the applicability of the presented method for the study of nonthermal kinetic effects in the distribution function. This can be rather open-ended, to initiate constructive discussion about the proposed method: estimates of dynamic timescales, angular extents and energy ranges would already allow the method to be scrutinized by experts specializing on specific phenomena.

Technical Corrections:

Simulated measurement plots (figures 1, 3, 5, 7, 8 and 9) are missing an axis label on

their (presumably) time axis.

References:

[1] Oñativia, J. & Dragotti, P. L. (2015). Sparse sampling: theory, methods and an application in neuroscience. Biological Cybernetics, 109, 125–139. doi: 10.1007/s00422-014-0639-x

[2] Greenaway, A. H. (1991). Optical aperture synthesis. Measurement Science and Technology, 2, 1.

[3] Kormann, K. & Sonnendrücker, E. (2016). Sparse Grids for the Vlasov–Poisson Equation. In J. Garcke & D. Pflüger (eds.), Sparse Grids and Applications - Stuttgart 2014 (p./pp. 163–190), Cham: Springer International Publishing. ISBN: 978-3-319-28262-6

[4] Pfau-Kempf, Y. P.-K., Battarbee, M., Ganse, U., Hoilijoki, S., Turc, L., von Alfthan, S., Vainio, R. & Palmroth, M. (2018). On the importance of spatial and velocity resolution in the hybrid-Vlasov modeling of collisionless shocks. Frontiers in Physics, . doi: 10.3389/fphy.2018.00044

[5] Leveque, R. (2002). Finite Volume Methods for Hyperbolic Problems. Cambridge, MA: Cambridge University Press. ISBN: 9780521009249

[6] Kajdič, P., Hietala, H. & Blanco-Cano, X. (2017). Different Types of Ion Populations Upstream of the 2013 October 8 Interplanetary Shock. The Astrophysical Journal Letters, 849, L27.
* * *

---

## Author Comment (AC1) · 7 Aug 2018

**Reply to Reviewer Comment #1**

The authors thank the reviewer for his/her thorough revision of the manuscript and the helpful comments. Below, we respond to each of the points that was raised.

**General Comments**

*Reviewer comment:*

```
- Table 1 and Section 2:
-- I would check to make sure that the shock jumps are correct,
```

[Figure]

```
as I recall from the CfA Shock Database that several shocks had
$\Delta$V > 200 km/s.
https://www.cfa.harvard.edu/shocks/wi_data/
-- You should reference some recent work that provides the first
long-term statistical study on solar wind parameters near 1 AU
by Wilson et al. [2018] (Note the supplemental material
does separate parameters by fast and slow wind).
-- I doubt either of these will modify the values in your table
very much, but they will provide at least a reference/source
for the provided values.
```

*Response:*
The reviewer is absolutely right in pointing out that some shocks at 1 AU involve $\Delta V > 200$ km/s. In formulating the instrument requirements, we do not require that the beam tracking algorithm should be able to capture all shocks completely, but most of them. The $< 200$ km/s" should therefore be read as "most of the time". Checking the publication mentioned by the reviewer indeed confirms the values that we list in the table.

*Modifications in text:*
We have added a footnote in the table to point out that the values for the shock $\Delta$V are "most of the time" and refer there to the CfA shock list. We have added the reference suggested by the reviewer regarding typical solar wind parameters in section 2.

*Reviewer comment:*

```
- Section 2.1: [The following are my musings, but are most
likely not critical]
-- I see you addressed most of my concerns below in Section 3
already, but I leave it here for reference.
```

-- One thing of which to be careful are secondary/reflected
ions near strong collisionless shocks. I assume you have
thought of this and know how to handle it, but I should
mention that even when the reflected to incident ion density
is relatively low, it can affect the bulk flow velocity
estimate determined from typical velocity moment software
significantly.
If the spacecraft on which the instrument of interest in
this paper is to orbit Earth and not, say, L1, then bow
shock reflected ions will be an issue and the fraction of
reflected-to-incident is much higher (>25% in some cases)
than typical interplanetary shocks.
This can affect the bulk flow velocity causing it to devaiate
away from the core solar wind proton beam by upwards of 30%,
i.e., >100 km/s [e.g., Wilson et al., 2014a]. In the case of
a sun-pointed spinner on an outbound pass, the number of
reflected ions entering the detector will likely be small,
so probably not an issue. However, the reflected ions at
earthward propagating interplanetary shocks will always be
an issue. The primary difference is that most interplanetary
shocks do not reflect a significant enough fraction of the
upstream ions to generate much of a foreshock, so perhaps
this is not cause for concern?

*Response:*
When restricting an instrument's field of view to a cone around the solar wind direction,
it is obvious that one cannot measure the reflected ions. The idea – as originally
foreseen on THOR – is to have both a fast beam tracking solar wind spectrometer and
an omnidirectional spectrometer (slower, offering some mass separation capability)

operating concurrently. For THOR also the goal was to go well out into the solar wind, so as to be sure that measurements are not affected by the foreshock. Reflection from solar wind shocks indeed remains an issue.

*Modifications in text:*
In the conclusions, we have inserted a paragraph discussing the utility of combining a beam tracking instrument with an omnidirectional spectrometer.

*Reviewer comment:*

```
-- I know of at least one interplanetary shock that caused
problems for the PESA Low detector from Wind/3DP that was
seen on 2001-11-24 near 05:51 UT. The thermal energies got
so large that the instrument lost the solar wind beam and
did not enter tracking mode because it thought it was still
following the beam. Granted, the mode was not as well designed
as newer spacecraft that use NV (i.e., roughly the count rate)
but it is worth considering.
```

*Response:*
This confirms the importance of a robust beam loss recovery strategy!

*Reviewer comment:*

```
-Section 3.4:
-- Be careful with the estimates of the spatial scales for
discontinuities. The thickness of the shock ramp is not on
ion scales, but on electron scales [e.g., Hobara et al., 2010;
Mazelle et al., 2010]. What is not shown in the Spektr-R data
is what was assumed for years to be the actual shock ramp but
was undersampled [e.g., see Wilson et al., 2012, 2017].
```

In general, I think your estimates are fine, but the statement
that ion properties cannot change faster than ion scales is
factually incorrect. Further, it is not the case that the
fluctuations discussed in the above references have no effect
on the ions, as shown by Goncharov et al. [2014].

*Response:*
Agreed. The ion gyroradius is a characteristic of the spatial scale of variation of the ion
VDFs, but the scale can be smaller if the magnetic field changes more rapidly and/or if
there are strong localized electric fields – and there the electron scales can come into
play.

*Modifications in text:*
We have reformulated this paragraph, and refer to Mazelle et al. (2010) and Krasnosel-
skikh et al. (2013) who discuss spatial scales in shocks.

*Reviewer comment:*

- Section 4.1
-- I am confused. If you have a sun-pointed spinning spacecraft
and you align the central elevation angle bin with roughly the
Earth-sun line, why does the solar wind beam vary with spin in
the elevation angle? Or am I misunderstanding Figure 1 and the
discussion in this section? Is the spacecraft spin axis not
aligned with the Earth-sun line?

*Response:*
The goal here is to illustrate what happens if the solar wind arrival direction does not
coincide with the spacecraft spin axis. That is going to happen very often. There is the
solar wind aberration angle that changes continuously within a range of a few degrees.

But it is also very unlikely that the spacecraft spin axis not aligned with the Earth-sun line. Indeed, suppose the alignment is perfect at a given instant, it will be 360°/365.25 = 1° off one day later because the spinning spacecraft axis keeps a constant direction in an inertial frame. Spacecraft operators would not want to do manoeuvres to reorient the axis on a daily basis (and the scientists wouldn't like that either).

*Modifications in text:*
We do think the explanation in 4.1 is clear enough.

*Reviewer comment:*

```
-- Page 7, Lines 27-30: I do not follow the sentence starting
with "The difference between..." Is this a comment on the results
shown in Figure 1 or a general comment about the solar wind?
```

*Response:*
That is a comment regarding the results. We simply want to point out that the measured arrival direction matches quite closely the true values with which we have set up the simulation.

*Modifications in text:*
We have adapted the phrase for clarity.

*Reviewer comment:*

```
-- Page 8, Lines 4-5: Can you be a little more quantitative
with the statement "...distributions are somewhat distorted..."?
Distorted in what way? Would one interpret the VDFs as having a
higher temperature than reality, for instance? If so, by how much?
```

*Response:*
The errors in arrival direction are quantified in Figure 2. The VDF distortion is illustrated

in Figure 3. "Rotational smearing" of the VDFs will not affect the mean energy that is measured, but it affects the mean arrival direction angles and it leads to a temperature anisotropy. Such high spacecraft spin rates are undesirable anyhow and one should stay away of that regime.

*Modifications in text:*
We inserted a phrase to describe the nature of the distortion more clearly (but still rely on Figure 3 to illustrate it).

*Reviewer comment:*

```
- Section 4.3
-- Having had several long conversations with Drs. Safrankova
and Nemecek (a few years ago now) about the capabilities and
limitations of the BMSW instrument, I am curious how you
managed to get the data into GSE coordinates. It was my
understanding that there is no way to know the actual spacecraft
orientation and attitude necessary to rotate the data out of
spacecraft coordinates into a physically meaningful basis.
Has this issue been recently resolved?
```

*Response:*
The instrument is mounted on the solar panels which can rotate. The exact solar panel rotation angle is not always known, which renders it impossible to derive the exact instrument look direction. However, for a considerable fraction of the time, including the events considered here, the solar panel rotation angle is fortunately available (though at a limited time resolution) and so the data can effectively be rotated into the GSE frame. We are particularly thankful to the referee for asking this question: digging deeper into this matter, we found out that we had actually NOT used the data in the GSE frame, but in the instrument frame, which, for the shock event, had its x-axis pointing about $11°$ away from the sun.

*Modifications in text:*
We have rerun the simulations for examples 7-8-9 using the data in the GSE frame and we have updated the figures. Note that, while we had originally observed that the solar wind seemed to go out of the CSW field-of-view, this now no longer is the case – this was simply due to the off-pointing x-axis. In retrospect, this should have triggered us to be suspicious of the reference frame of the original data. We have made the corresponding modifications in the text where we discuss these simulations. The paragraph in the conclusions that commented on the CSW field-of-view was also adapted.

*Reviewer comment:*

```
-- The shock on 2015-06-22 arrived at L1 at ~18:08:24 UT (e.g.,
I looked at Wind data on CDAWeb). Regardless, the bulk flow
velocity along X-GSE jumps to nearly -800 km/s in the
downstream and the ion temperature exceeds 100 eV (i.e., ~1.2 MK),
so the temperatures may not be too inaccurate from BMSW. The CfA
shock database shows a density compression ratio of ~3.4 but I
think the temperature changes by a factor >4-5. [These are just
comments, not really actionable items.]
-- Page 9, Lines 50-51: Are the temperature and temperature
anisotropy significantly affected as well, or just the density
moment?
```

*Response:*
Thanks for checking this shock with the Wind data. It can indeed be interesting to try to compare some of the BMSW data with shock measurements elsewhere in geospace. As stated in the text, the temperature measurement is affected too. BMSW does not provide temperature anisotropy.

[Figure]

*Reviewer comment:*

```
- Hot and/or Tenuous VDFs
-- One of the biggest issues that I did not see addressed in
the manuscript occurs during intervals when the density is low
[i.e., below ~1 cm^(-3)] or the temperature is high (i.e.,
Ti > ~100-200 eV, depending on the instrument). If we assume
a bi-Maxwellian or even an isotropic Maxwellian, the peak phase
space density goes as N*T^(-3/2). The one-count level
during the same interval does not drop/change relative to an
adjacent, earlier interval. Thus, the signal-to-noise ratio
can drop preciptously during these periods. I realize this
is an issue faced by all particle instruments, but it is
worth discussing to ensure you do not lose the critical parts
of the distribution downstream of strong shocks with high
temperatures but relatively low density (e.g., for really
low upstream density).
```

*Response:*

The referee is absolutely right in stressing the importance of making sure that there are no problems with the signal-to-noise ratio. We want to point out 3 elements in this respect:

- As mentioned in the introduction, any plasma spectrometer faces a trade-off between (a) angular and energy resolution, (b) time resolution, (c) signal-to-noise ratio. Obviously this trade-off is linked to hardware limitations (e.g. the instrument's geometrical factor is limited by the volume and mass budget, there are constraints due to the telemetry budget, etc. . .). It is precisely here that beam tracking is useful: by making measurements only where it matters, the best trade-off remains possible. For instance, for given time, angular and energy resolutions,

beam tracking allows to maximize the data collection time per measurement bin so that even for low count rates a significant number of counts can be collected, thereby improving the signal-to-noise ratio. So implementing beam tracking in general helps to avoid low counts.

- The important question here is whether the beam tracking strategy would not get confused in low density / high temperature environments. With the simple beam loss detection strategy used here, low densities would trigger the "beam loss" condition. But that would not be dramatic: the instrument simply returns to a measurement strategy that samples the full phase space accessible by the instrument. Although one would lose time resolution, providing VDFs over the full phase space is one of the best things one can do in such a situation (especially for the high temperature case). A posteriori, one can bin the measurements in energy, azimuth, elevation and/or time to improve the signal-to-noise ratio even further so that these measurements are scientifically useful.

- Beam tracking driven by a Faraday cup instrument would suffer less from problems in such situations, since a Faraday cup inherently provides a better signal-to-noise as it integrates the particle flux over its entire field of view.

*Modifications in text:*
We have inserted a paragraph in section 3.3 (Beam loss detection and recovery) discussing this matter.

**Minor Concerns**

*Reviewer comment:*

```
-- Page 1, Lines 35-50: You could also mention waves and
instabilities [e.g., Malaspina et al., 2013], as
electromagnetic fluctuations are not solely limited to
```

```
turbulence. It is also important to measure the full 3D
VDFs for analysis of instabilities.
```

*Modifications in text:*
Sure. We have added a sentence + a few references.

*Reviewer comment:* – Page 2, Lines 2-18: The Wind spacecraft's 3DP instrument suite is also relevant here [e.g., Lin et al., 1995].

*Modifications in text:*
We have added a sentence about 3DP (mentioning also its higher angular resolution near the ecliptic plane) as well as the reference.

*Reviewer comment:*

```
-- Page 2, Line 47: I know voxel is a term analogous to a
velocity-space pixel, but could you provide a definition
for the reader that may not know this.
```

*Modifications in text:*
Provided a definition upon first occurrence

*Reviewer comment:*

```
-- Page 7, Lines 10-12: I am not sure I understand the
sentence starting with "It starts measuring..." You state
the instrument starts sampling at 600 ms and the duration
required to obtain one full VDF is another 600 ms. Is that
correct?
```

*Response:*
Yes, absolutely correct.

**Typos, Grammar, etc.**

*Reviewer comment:*

```
[The following are suggestions, not requirements
(e.g., I do not recall rules for British vs.
American grammar for when to use commas after things
like "e.g." or "i.e.")]
Page 4, Line 25: "12, i.e. an order" --> "12, i.e., an order"
Page 5, Line 56: "i.e. one uses" --> "i.e., one uses"
Page 5, Lines 77-79: "In order to eliminate values that are
completely off, a voting" -->
"In order to eliminate outliers, a voting"
Page 5, Line 87: Try rephrasing the following "Note that
such a more robust procedure
requires" as it is awkwardly phrased and not clear what is meant.
Page 6, Line 38: "robust (i.e. when" --> "robust (i.e., when"
Page 6, Line 40: "...cient (i.e. when" --> "...cient (i.e., when"
Page 6, Line 98: "direction (i.e. with" --> "direction (i.e., with"
Page 8, Lines 62-63: "The measurement points" --> "The measurements"
Page 9, Line 19: "neither dramatic in magnitude nor very" -->
"neither dramatic in magnitude or very"
Page 11, Line 5: "instrument (i.e. of" --> "instrument (i.e., of"
```

*Modifications in text:*
Thanks, all have been dealt with.

*Reviewer comment:*

```
Page 14, Line 5: "manoeuvres" --> "maneuvers"
```

*Response:*
We stick with British English.

**References**

-- Goncharov, O., et al., "Upstream and downstream wave
packets associated with low-Mach number interplanetary shocks,"
Geophys. Res. Lett. 41, pp. 8100---8106,
doi:10.1002/2014GL062149, 2014.
-- Hobara, Y., et al., "Statistical study of the quasi-
perpendicular shock ramp widths," J. Geophys. Res. Vol. 115,
pp. A11106, doi:10.1029/2010JA015659, 2010.
-- Lin, R.P., et al., "A Three-Dimensional Plasma and
Energetic Particle Investigation for the Wind Spacecraft,"
Space Sci. Rev. Vol. 71(1), pp. 125--153,
doi:10.1007/BF00751328, 1995.
-- Malaspina, D.M., et al., "Electrostatic Solitary Waves
in the Solar Wind: Evidence for Instability at Solar Wind
Current Sheets," J. Geophys. Res. Vol. 118, pp. 591—599,
doi:10.1002/jgra.50102, 2013.
-- Mazelle, C., et al., "Self-Reformation of the Quasi-
Perpendicular Shock: CLUSTER Observations," Proc. 12th
Int. Solar Wind Conf., AIP Conf. Proc. 1216, pp. 471--474,
doi:10.1063/1.3395905, 2010.
-- Wilson III, L.B., et al., "Observations of
electromagnetic whistler precursors at
supercritical interplanetary shocks," Geophys. Res. Lett.
Vol. 39, L08109, doi:10.1029/2012GL051581, 2012.
-- Wilson III, L.B., et al., "Quantified energy dissipation

rates in the terrestrial bow shock: 1. Analysis techniques
and methodology," J. Geophys. Res. Vol. 119, pp. 6455--6474,
doi:10.1002/2014JA019929, 2014a.
-- Wilson III, L.B., et al., "Revisiting the structure of
low-Mach number, low-beta, quasi-perpendicular shocks,"
J. Geophys. Res. Vol. 122, pp. 9115--9133,
doi:10.1002/2017JA024352, 2017.
-- Wilson III, L.B., et al., "The Statistical Properties
of Solar Wind Temperature Parameters Near 1 au," Astrophys.
J. Suppl. Vol. 236(2), pp. 41, doi:10.3847/1538-4365/aab71c,
2018.

---

## Author Comment (AC2) · 7 Aug 2018

**Reply to Reviewer Comment #2**

The authors thank the reviewer for his/her thorough revision of the manuscript and the helpful comments. Below, we respond to each of the points that was raised.

**General Comments**

*Reviewer comment:*

The presented manuscript presents and discusses a novel approach to employ electrostatic spacecraft analyzers

fitted with angular deflectors. By evaluating beam
parameters of the surrounding plasma, only energy-
and directional bins relevant to resolving said beam
need to be sampled, resulting in much faster signal
acquisition and as a result, higher time resolution.

The presented method represents an instance of a sparse
sampling approach, in which the sample points from a
high-dimensional parameters space are deliberately
constrained to certain subsets of that space in order
to obtain a maximum amount of information with minimal
sampling  requirements. Similar techniques have been
employed with great success in Biophysics (Such as
compressed sensing techniques in neurosciences [1]),
Astronomy (in aperture synthesis for telescopes [2]).
Likewise in the same field as this manuscript, kinetic
simulation approaches in space physics employ similar
techniques to reduce the computational load of high-
dimensional simulation spaces [3,4].

References to similar approaches from those fields,
as well as overview papers of compressed sensing
methods should be added, since a large body of
general theoretical background work from other
fields can be applied for this approach.

Specifically, the presented manuscript discusses a
method to sparsely sample space plasma velocity
distributions, with the intention of tracking a

Interactive
comment
```
"beam" and sampling it with a minimum number of
required samples, to obtain an extraordinarily
high temporal resolution.
```

*Response:*
Indeed, beam tracking tries to represent a system with a maximum amount of infor-
mation for a minimum sampling effort. That there is a general theoretical framework
regarding "sparse sampling" is undisputable. However, we feel somewhat reluctant
to expand the manuscript with a discussion of the "sparse sampling" or "compressed
sensing" context for two kinds of reasons.

First: How relevant is "sparse sampling" or "compressed sensing" in this context?

- The "compressed sensing problem" deals with reconstructing a sparse vector
  from a reduced number of data with sparsity as a priori knowledge. The idea
  then is that measuring the limited data is sufficient to reconstruct the full data if
  one knows the sparsity properties. The practical relevance of this is very much
  dependent on the specific assumptions. "Beam tracking" is a very specific form of
  sampling, in which knowledge about smoothness and compactness of the VDFs
  in velocity space, and physical knowledge about the time scales involved, are
  all fundamental. In other words: the peculiarities of the situation at hand are
  responsible for the fact that one cannot learn very much from the generic theory.
  To put it simply: We measure data in a compact subregion of phase space (the
  reduced data), and from that we derive the full data by simply assuming that the
  VDF is zero outside that subregion (finding the full data from the known sparsity
  pattern). This sort of application is so trivial that we do not need the general
  theory. Of course, more refined approaches could be possible in which one might
  sample a few isolated points in phase space to derive the full VDF from that, but
  this would be strongly dependent on specific assumptions (for instance, you could
  do this efficiently if you assume that the distribution is a bi-maxwellian). However,

the scientists usually do not want to make those assumptions. Alternatively, one might train a subsampling algorithm on a set of realistic data . . . but we do not have such training data – we have at present no high time cadence VDFs.

- The standard "compressed sensing" theory does not take into account the notion that there might be variable costs associated with collecting the reduced data. There is a cost (a time delay) associated with switching the spectrometer to a different energy (due to the need to set high voltages and the accompanying settling times). It makes little sense to measure at a specific elevation angle, since it is much more time-efficient to sweep the voltage between the deflector plates to acquire a contiguous set of measurements over all elevations at once. All these practical constraints limit the amount of sparsity in the problem, so that there is little to gain from the general approach.

- Compressed sensing has a lot in common with data compression, in particular with lossy compression. That is something that scientists prefer to avoid. To really understand the measurements, and to be able to reprocess data, they prefer not to work with an "indirect" representation of the sampled velocity space (the reduced data set); they simply want to know the values as measured in all the individual voxels, not any reconstructed values. Indeed, it would be very awkward (though not impossible) to update an instrument calibration and reprocess the raw data if only an indirect representation is given.

- An additional problem with an "indirect representation" is that any further statistical analysis of the data will be hampered by the strong and specific cross-correlations between the errors on the measurements in different voxels.

- For beam tracking and moment calculation, one must be able to interpret the measured data on-board in a straightforward manner and fast. It is not at all clear whether the typical optimization techniques used in compressed sensing and
the required response times are within the capabilities of present-day on-board processors.

Second: How relevant is it to mention "sparse sampling" for the reader of the paper?

- Our introduction briefly reviews a number of plasma spectrometers, showing the progress of technology and actually focusing on the way in which the velocity space sampling problem has been approached. That sketches the context for the discussion of beam tracking sufficiently well to allow the reader to follow the text. We are therefore not convinced of the necessity of inserting an overview of sparse sampling techniques in the introduction.

- We want to point out that every measuring instrument is doing one form or another of sparse sampling, yet descriptions of instruments typically do not mention sparse sampling theory (none of the reference papers for the plasma spectrometers that are reviewed in the introduction does). Admittedly, it is not because nobody does it, that it could not be useful.

- We are already a bit concerned about the manuscript length and do not want to expand it unnecessarily.

*Modifications in text:*
Given all the above, we have inserted a paragraph that mentions the possibility to interpret beam tracking in the context of "sparse sampling" or "compressed sensing", without entering into a deeper discussion that would necessitate to mention some of the points listed above. We have chosen to do so in the discussion section, rather than in the introduction. This allows us to present sparse sampling methods as a possible future avenue in the quest for even faster solar wind characterization. We have added a few general references, and one targeted toward the space plasma physics audience.

*Reviewer comment:*

```
The model assumptions going into the example analysis
performed in this manuscript are a) that the "interesting"
part of the particle distribution is quite compact in
shape, more precisely, in this analysis it is assumed
to be maxwellian b) that it's overall shape stays the
same, and only it's parameters change. These assumptions
preclude the possibility of multiple mixed plasma
distributions, such as a core and beam setup in a
foreshock, rings or loss cones in a fermi-type
acceleration region or any other non thermally-relaxed
particle distribution.

I assume that the authors only focus on the solar wind
distributions' core is motivated by their specific
research interests. However, the study of kinetic
physics of the solar wind, including the effects of
turbulence, shocks and magnetic reconnection depends
strongly on the ability to study and understand
nonthermal distribution functions, that is, precisely
those distribution functions that do not fulfill the
assumptions going into the manuscript at hand.
```

*Response:*
There seems to be a serious misunderstanding here. We make no assumption regarding the shape of the velocity distribution function other than that it is compact, i.e., that it occupies only part of the phase space accessible by the instrument. We use Maxwellians only to test the beam tracking algorithm, since we do not possess any VDF measurements at the high cadence considered here. The shape of the distribution

is allowed to change, it can be anisotropic, it can consist of a core and halo, it can be a mixture of different populations, the particle distribution does not have to be thermally relaxed . . . as long as the compactness condition is satisfied. The degree to which this must be the case depends on the choice of the phase space energy/azimuth/elevation window sizes. Obviously, this approach cannot deal with non-compact distributions. It precludes, for instance, the detection of backstreaming solar wind particles reflected from an interplanetary shock: an instrument staring at the sun cannot detect particles coming from behind.

*Modifications in text:*
Also in view of a comment by the other referee, we have inserted a paragraph in the conclusions that addresses the issue of populations reflected from shocks.

*Reviewer comment:*

```
While a much more thorough analysis and quantification
of the detector behavior for realistic distribution
functions will be required before the presented method
can be employed in an actual instrument, it is probably
not within the scope of this paper to perform them --
the central subject and conclusion being the presentation
and motivation of a sparse sampling scheme in the first
place. Still, some more reflection on the limitations of
the presented analysis, and avenues to further refine
the analysis should be included.

In conclusion, this manuscript presents a thoroughly
novel idea that merits publication and discussion in
the wider scientific community, but suffers from being
too narrow in it's goals and scope. After some major
revisions, in which the presented method is evaluated
```

```
with a focus on more general kinetic-physics processes,
I consider it suitable for publication.
```

*Response:*
We're a bit surprised that the reviewer asks us to evaluate the method with a focus on general kinetic physics requirements, as we believe that the present manuscript does exactly that. Indeed, studies of turbulence, waves, or instabilities require only two things of a plasma spectrometer: (a) obtain VDFs with high energy and angular resolution, and (b) with high time resolution. That the first goal can be achieved with the THOR-CSW design parameters, has been demonstrated by Valentini et al. (2016) as cited in the manuscript. The present study of beam tracking demonstrates that the first goal can be achieved while at the same time satisfying the second goal. That demonstration consists of

- Showing that with realistic sizes of the beam tracking window and the prescribed angular and energy resolutions, a high time resolution can be achieved.

- Showing that – and this is what we perceived as being the major concern – the beam tracking technique is able to deal with dramatic time evolution in the solar wind, as exemplified by shocks. There is no reason to worry about rapid, but less dramatic, changes at or near the centre of the solar wind beam: they will be captured as long as the beam tracking window is large enough.

The description of how beam tracking can be implemented, as well as the examples, corroborate our claim that beam tracking is indeed capable of leading to solar wind VDF measurements that enable (ion) kinetics-physics studies.

**Specific Comments**

*Reviewer comment:*

The prediction method presented in section 3.2 and it's
discussion of polynomial extrapolation overshoots is
very similar in nature to the problem of flux limiters
in finite volume simulation methods, such as MHD simulation.
There, too, the extrapolation of a reconstruction polynomial
is clamped to remain within physically realistic boundaries.
This similarity could be discussed and referenced
(such as [5]).

*Response:*

There is indeed a certain similarity with flux limiters used in computational fluid dynamics, but the setup is different: in the CFD case, one knows the gradients at both sides of an interface, and the ratio between those two gradients is used as the argument of the flux limiter to bound the extrapolations from either side. In the case of beam tracking, one only knows the gradient from the past; nothing is known about the future. Actually, there are numerous other situations where an extrapolation (which is always risky) can be bounded by using additional heuristic knowledge. Stock market prediction is actually much more similar to the beam tracking problem that we are dealing with. However, we think that making that comparison in the manuscript would lead us too far astray.

*Reviewer comment:*

The same section claims that "All in all, one can expect
such techniques to work reasonably well only if the energy
does not change rapidly", and I agree with that statement.
However, especially in shocks, discontinuities and
reconnection regions, where this assumption does not
hold true, is where the most interesting kinetic plasma
physics effects occur.

*Response:*
This sentence refers specifically to the use of higher order polynomial interpolation. The point that we make here is that linear interpolation is actually better in view of shocks or discontinuities, as explained in the preceding sentence.

*Modifications in text:*
We have rephrased the sentence to avoid any misunderstanding.

*Reviewer comment:*

```
Note that the sudden changes of distribution function in
these events are not simply a parameter change of a
maxwellian: the shape of the distribution function
departs *significantly* from a maxwellian whenever
kinetic physics comes into play. If the spacecraft
changes it's magnetic connection to a shock, beam
distributions of highly nonthermal shape can suddenly
"appear" outside of the thermal velocity radius of the
previous maxwellian. In reconnection regions, spitzer
orbits and crescent-shaped velocity distributions
additionally appear on top of any thermal background
that might still be present. Additionally, nonisotropic
superthermal tails can deform the solar wind distribution
away from a maxwell-boltzmann shape.
```

*Response:*
We reiterate that we make no assumption regarding the shape of the velocity distribution function that has to be measured, although our tests are limited to Maxwellian distributions. We understand very well the risks of monitoring only a limited part of phase space – indeed, one can miss certain features. This is why a beam tracking

plasma spectrometer is best used in combination with an omnidirectional spectrometer, where the former gives you very high time resolution for the core of the distribution, and the latter provides the full context (but probably at a slower pace).

*Modifications in text:*
A paragraph inserted in the conclusions addresses this complementarity of both types of spectrometer.

*Reviewer comment:*

```
The discussion in section 4.4, comparing internal and
external beam tracking, is thus incomplete, as the
asumption of a continuous change of maxwell distribution
parameters won't represent reality in many interesting
kinetic physics scenarios.
```

*Response:*
Maxwellian distributions are used here only to construct a test example. The emphasis of the test in section 4.4 is on achieving the necessary time resolution and being able to follow the rapid velocity jumps (jumps in energy and/or beam direction). The reviewer is of course correct in that the changing nature of the distributions themselves could play a role as well; in that sense the test is indeed incomplete. However, nobody has ever measured distributions that rapidly, so we had to construct the example artificially. An alternative could have been to use the output of a full Vlasov simulation code, but we do not have access to such simulations over a sufficiently long time period and featuring solar wind shocks. It is our opinion that "the proof of the pudding is in the eating", i.e., a full evaluation of beam tracking is only possible by building an instrument and trying it out in space.

*Modifications in text:*
We have inserted a short discussion concerning the incompleteness of the test as the

Interactive
comment

last paragraph in section 4.3 (this issue is applicable to all the tests with the BMSW data, and goes beyond the internal/external beam tracking differences).

*Reviewer comment:*

```
As for the beam loss criterion itself (sections 3.3 and
3.4), it is based on the assumption that the "beam"
encompasses the entire interesting part of the distribution
function at time of tracking, and that the only noteworthy
change at a plasma discontinuity would be a sudden loss
of the beam at one spot, with reappearance at another.
This is a rough oversimplification of the wide variety
of foreshock distribution functions (compare [6]): in
many cases, additional beam distributions will occur
far outside the thermal velocity extents of the solar
wind beam, thus remaining untracked by the restricted
sampling process presented here. "Beam Loss" as defined
in this paper is neither an appropriate, nor a sufficient
criterion for re-scanning of the complete velocity space.
```

*Response:*
No, the beam tracking spectrometer can follow all sorts of changes in the shape of the distribution functions across a discontinuity, as long as these occur not too far away from the centre of the beam. We agree that this condition is not valid in the foreshock. As mentioned before, this is why a beam tracking plasma spectrometer is best used in combination with an omnidirectional spectrometer.

*Modifications in text:*
This is now addressed in the conclusions section.

*Reviewer comment:*
```
The presented tests inadequately asses the response of
```

Interactive
comment
```
the method to these kind of scenarios. While it is good
and correct to assess the ability of this method to re-
acquire the beam after a beam loss scenario with realistic
dynamic timescales, this is, by far, not the only relevant
measurement quantity to optimize for. I would suggest
expanding section 4 with a discussion of the applicability
of the presented method for the study of nonthermal
kinetic effects in the distribution function. This can
be rather open-ended, to initiate constructive discussion
about the proposed method: estimates of dynamic timescales,
angular extents and energy ranges would already allow the
method to be scrutinized by experts specializing on
specific phenomena.
```

*Response:*
What the reviewer proposes here goes far beyond the demonstration of beam tracking as a viable method to operate a plasma spectrometer so as to satisfy the generic requirements for doing kinetic-scale physics. A detailed examination of what can be done by a specific spectrometer with given energy/azimuth/angular and time resolutions for a particular set of non-thermal kinetic effects (as simulated by a Vlasov code) could indeed be the subject of a follow-up paper.

**Technical Corrections**

*Reviewer comment:*

```
Simulated measurement plots (figures 1, 3, 5, 7, 8 and 9)
are missing an axis label on their (presumably) time axis.
```

*Modifications in text:*
We added the axis label on all those plots.

**References**

[1] O\~nativia, J. & Dragotti, P. L. (2015). Sparse sampling: theory, methods and an application in neuroscience. Biological Cybernetics, 109, 125--139. doi: 10.1007/s00422-014-0639-x

[2] Greenaway, A. H. (1991). Optical aperture synthesis. Measurement Science and Technology, 2, 1.

[3] Kormann, K. & Sonnendr\"ucker, E. (2016). Sparse Grids for the Vlasov--Poisson Equation. In J. Garcke & D. Pfl\"uger (eds.), Sparse Grids and Applications -- Stuttgart 2014 (p./pp. 163--190), Cham: Springer International Publishing. ISBN: 978-3-319-28262-6

[4] Pfau-Kempf, Y. P.-K., Battarbee, M., Ganse, U., Hoilijoki, S., Turc, L., von Alfthan, S., Vainio, R. & Palmroth, M. (2018). On the importance of spatial and velocity resolution in the hybrid-Vlasov modeling of collisionless shocks. Frontiers in Physics, doi:10.3389/fphy.2018.00044

[5] Leveque, R. (2002). Finite Volume Methods for Hyperbolic Problems. Cambridge, MA: Cambridge University Press. ISBN: 9780521009249

[6] KajdiËǦc, P., Hietala, H. & Blanco-Cano, X. (2017). Different Types of Ion Populations Upstream of the 2013 October 8 Interplanetary Shock. The Astrophysical Journal Letters, 849, L27.

---

## Referee Report (RR1)

The authors' response has adressed my concerns with the presented analysis. In particular, the paragraphs added in relation to the solar wind tracking and conclusions sections help greatly with defusing my criticism towards the choice of test model employed here: Through the way the manuscript was formulated upon first submission, I had gotten the impression that the authors are advocating for the beam-tracking detector by itself as completely sufficient for any kind of study of the particle distribution function in the solar wind. The added text illuminates some of the limitations of the beam tracking process, and illustrates its specific place in combination with other devices while still comprehensively outlining the benefits such an instrument provides for solar wind measurements.

As a result, the choice of simulation analysis presented now makes a lot more sense to me: the restriction to maxwellian distribution functions is entirely sufficient to test the performance of an instrument that is designed to study the behaviour of compact velocity space distributions.

Mention of sparse sampling methods and references to corresponding literature have been added, and while they are fewer and smaller in scope that I had suggested, the authors convincingly argued that more extended treatment of sparse sampling would go beyond the scope of this manuscript. Thus, I consider it sufficient to educate potential readers about the subject.

I thank the authors for their thorough and enlightening response to my comments, and now recommend the paper for publication.